# The Emergence of Complex Behavior in Large-Scale Ecological Environments

## Abstract

We explore how physical scale and population size shape the emergence of complex behaviors in open-ended ecological environments. In our setting, agents are unsupervised and have no explicit rewards or learning objectives but instead evolve over time according to reproduction, mutation, and selection. As they act, agents also shape their environment and the population around them in an ongoing dynamic ecology. Our goal is not to optimize a single high-performance policy, but instead to examine how behaviors emerge and evolve across large populations due to natural competition and environmental pressures. We use modern hardware along with a new multi-agent simulator to scale the environment and population to sizes much larger than previously attempted, reaching populations of over 60,000 agents, each with their own evolved neural network policy. We identify various emergent behaviors such as long-range resource extraction, vision-based foraging, and predation that arise under competitive and survival pressures. We examine how sensing modalities and environmental scale affect the emergence of these behaviors and find that some of them appear only in sufficiently large environments and populations, and that larger scales increase the stability and consistency of these emergent behaviors. While there is a rich history of research in evolutionary settings, our scaling results on modern hardware provide promising new directions to explore ecology as an instrument of machine learning in an era of increasingly abundant computational resources and efficient machine frameworks. Experimental code is available at **url withheld to preserve anonymity.**

## 1 Introduction

Since Darwin, the evolutionary emergence of biodiversity and complex behavior has been studied across a wide variety of biological disciplines, including evolutionary theory (Darwin, 1859; Simpson, 1944), ecology (MacArthur & Wilson, 2001), and genetics (Dobzhansky, 1937). Models of these emergent dynamics range from theories of speciation and adaptive radiation (Coyne & Orr, 2004; Schluter, 2000) to the mathematical analyses of population genetics (Fisher, 1930; Wright, 1931).

In these fields, decades of mathematical modeling, laboratory experiments, and field work have tremendously improved our understanding of the mechanisms of evolution in nature. Unfortunately, there are many obstacles to studying emergent behavior in the natural world. As ecosystems become larger and more complex, they are more difficult to control and measure. Even where possible, running controlled experiments in large-scale natural settings risks damaging or displacing wild populations.

In an effort to better understand the emergence of complex behavior due to competitive and environmental pressures, we study the open-ended evolution of neural network policies in large-scale ecological simulations. In this setting, agents do not have a specified objective or reward signal but instead collect resources to survive and reproduce according to the dynamics of the environment. Policy changes occur only via mutation as parents pass their policies on to their children. With the rapid growth in computational resources available to researchers, our goal is to study open-ended evolutionary dynamics in populations of these simple digital organisms at scales not previously possible, in the hopes of providing new tools for investigating the effects of environmental scale and population size on behavioral emergence.

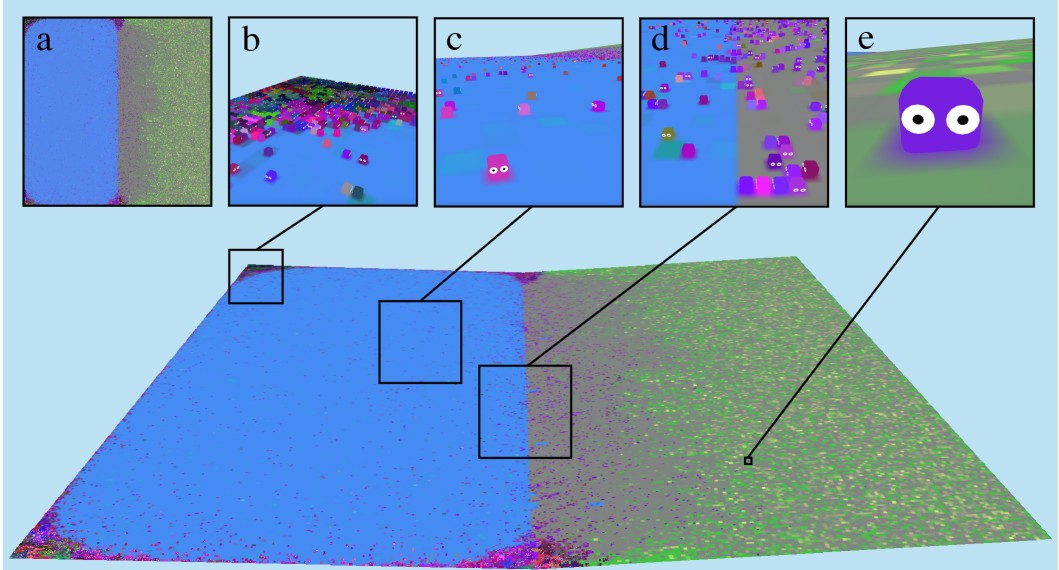

Figure 1: A 3D visualization of a $512 \times 512$ map in our environment. Insert (a) shows a top down view of the entire map, (b) shows a group of agents clustered in the corner, (c) shows agents in the water, (d) shows agents travelling back and forth from the land and (e) shows a close up of a single agent gathering resources.

To study this, we use a new JAX-based simulation environment that allows rapid evaluation of large grid worlds. Our larger experiments can contain over 60,000 individual agents with a physical area of over 1,000,000 grid cells. Agents in this environment must navigate to find food and other resources in order to survive. Agents can also reproduce by collecting enough resources to make a mutated copy of themselves. A 3D visualization of this environment can be found in Figure 1.

Our goal is not only to better understand ecological phenomena through simulation, but to also study ecology as a mechanism for producing machine intelligence. Recent years have demonstrated that in AI, scale doesn't merely improve models—it fundamentally transforms their capabilities (Wei et al., 2022a). Yet despite this insight, many attempts at embodied learning have been often been confined to relatively small environments with low populations. Our work explores what capabilities emerge when we scale not just a single model but an entire ecosystem. Just as reasoning emerges in individual models only beyond a certain size threshold (Wei et al., 2022b), we seek to understand what strategies evolve within populations only when environmental size and complexity reach sufficient richness.

Through extensive experiments, we identify multiple instances where sensor configurations and our large environmental scales strongly impact the behaviors that emerge from simulation, which can lead to long-term ecological consequences. For example, we show that agents with simple compass sensors can adapt to conduct long-range resource gathering expeditions, but that this behavior emerges less reliably at small scales. We also show that agents with vision sensors will evolve better foraging and attacking behavior than agents without them. Again, these effects appear more reliably at large scales. These experiments demonstrate that the scale and complexity enabled by modern hardware accelerators offer promising new approaches to study how evolution in ecological environments can produce rich new behaviors.

## 2 RELATED WORK

The emergence of life has inspired computational research since the field's inception. Conway's Game of Life (Games, 1970) simulates populations through simple local rules. Early computer programs such as Tierra (Ray, 1992) and Avida (Brown) simulate life as self-replicating programs

that evolve through mutations. Like Conway, Reynolds (1987) used local rules of motion to produce emergent flocking and schooling behaviors.

Researchers have used these principles to evolve the locomotion morphology (Sims, 1994a;b; Ha, 2018; Gupta et al., 2021; Auerbach et al., 2014; Auerbach & Bongard, 2012; Jeon et al., 2025; Auerbach & Bongard, 2014; Szerlip & Stanley, 2013) and sensing (Pratt et al., 2022; Mugan & MacIver, 2020; Tiwary et al., 2025) of virtual agents, while neuroevolution methods have evolved neural network architectures directly (Stanley & Miikkulainen, 2002; 2004; Stanley et al., 2009). In contrast to these approaches, we do not attempt to produce agents that can better adapt to their environment using some reward or objective signal, but instead seek to understand what behaviors naturally emerge as a result of ecological pressure and understand the impacts of environmental scale and complexity on these behaviors.

Evolution has also provided inspiration for the internal dynamics of optimization algorithms. Genetic algorithms (Holland, 1992; Jong, 1975) and evolutionary strategies (Wierstra et al., 2014; Hansen & Ostermeier, 2001; Rechenberg, 1973; Schwefel, 1995; Conti et al., 2018) apply the principles of evolution and natural selection to optimization. Other similar works use evolutionary methods to encourage novelty within the resulting solutions (Lehman & Stanley, 2011; Cully et al., 2015). Other works have used these approaches to optimize neural networks for tasks like Atari games and Mujoco locomotion tasks (Salimans et al., 2017; Such et al., 2017; Mania et al., 2018). While our work employs evolutionary mechanisms to update the agents, our focus is on studying emergent behavior rather than developing a novel optimization algorithm or using an existing one to accomplish a fixed objective.

There is also a rich history of work which takes biological inspiration to develop organisms within an environment. Early examples include Bedau (1992), Yaeger et al. (1994), Channon & Damper (1998), and Tisue et al. (2004). Some recent works use cellular automata to grow organisms with emergent morphologies and behaviors (Mordvintsev et al., 2020; Robinson, 2021; Heinemann, 2021; Randazzo & Mordvintsev, 2023; Caussan, 2025). Mordatch & Abbeel (2018) and Park et al. (2023) use simulated environments to investigate the emergence of language and communication between agents. The emergence of behavior due to autocurricula (Leibo et al., 2019), the idea that populations of agents provide an increasingly challenging learning landscape for each other, has also been explored in environments with smaller populations than ours (Leibo et al., 2018; Team et al., 2021). Additionally, Wang et al. (2019) and Aki et al. (2024) take the reverse approach, evolving environments specifically optimized for ideal agent learning. Though these works evolve the environment rather than the agent, they provide inspiration for our work as they highlight how environmental complexity can elicit sophisticated behaviors.

Most closely related to our work are several efforts that simulate populations of agents interacting within complex environments. Baker et al. (2019) examine a Hide-and-Seek task where a small number of hider agents learn to use environmental tools to evade seeker agents, both trained via reinforcement learning. Other works similarly train populations for specific tasks: playing video games (Jaderberg et al., 2019), manipulating objects (Petrenko et al., 2021), or capturing agents (Zheng et al., 2017; Yamada et al., 2020). However, while many of these works explore emergence, they do so using fixed tasks and objectives. The Neural MMO framework (Suarez et al., 2019; 2023) takes a more open-ended approach to reinforcement learning, placing agents within a massively multi-agent game setting and optimizing only for survival rather than fixed objectives. However, the population remains capped at a fixed number of agents (experiments examine up to 128 concurrent agents) without generational inheritance, limiting examination of population dynamics. Hamon et al. (2023) aligns more closely with our work through variable population sizes, genetically inspired evolution, and generational timelines, though its environment is significantly smaller, less feature-rich and it does not directly explore the importance of scale. Similarly, Lu et al. (2024) also simulates agents evolving to gather resources. However, this work also operates at a much smaller scale (a maximum of 256 agents) and emphasizes tool use through programmable robots. While all of these past works take steps towards examining emergent ecological dynamics, comprehensive understanding requires populations that can grow and shrink naturally, evolve without predefined objectives, and interact within environments rich enough to support diverse strategies at scales where collective patterns can emerge. Our work integrates these elements, enabling examination of ecological phenomena inaccessible in prior work.

Finally, we are also inspired by recent works that emphasize environmental speed and scale, using optimized libraries to increase the computational speed and number of agents able to interact in game-like environments (Koyamada et al., 2023; Matthews et al., 2024; Lechner et al., 2023). We build upon this progress in computational efficiency, utilizing similar JAX-based acceleration techniques to achieve large scales and fast simulation in ecological environments.

# 3 METHOD

## 3.1 ECOLOGICAL GAMES

We model the interactions between populations of agents and their environment as a type of multiplayer game we refer to as an **Ecological Game (EG)**. Similar to mathematical models such as Markov decision processes (MDPs) in reinforcement learning and stochastic games (SGs) in multi-agent reinforcement learning (see Sutton et al. (1998) and Albrecht et al. (2024) for reference), an EG contains underlying Markovian transition dynamics but is designed to express changing populations of agents in an objective-free setting. Like an SG, an EG contains a set of agents $I$, a state space $S$, an initial state distribution $\rho_0$, an action space $A$, and a transition function $\mathcal{T}$ mapping states $s_t$ and a set of actions for each player $a_{t,i}$ to a distribution over subsequent states $s_{t+1}$. Unlike an SG, it removes the reward function, and adds a population function $\Gamma(s_t)$ that provides information about which agents in $I$ are currently alive and the identity of their parents so that birth and death events can be inferred. Just as MDPs and SGs have partially observable counterparts, POMDPs and POSGs, an EG can also be partially observable, in which case we refer to it as a POEG. The difference between a POEG and an EG is that a POEG does not allow each agent to observe the entire state, and instead incorporates an observation function $O(s_t, i)$ that maps a state and player to an observation for that player. Our environment is a POEG with different observation functions depending on various sensor configurations explained in the next subsection.

## 3.2 ENVIRONMENT

The particular POEG used in our experiments is a large grid world containing various resources that agents must collect in order to reproduce and survive. The environment maintains a factored state space $s_t = \{s_t^m, s_t^a\} \in S = S^m \times S^a$ where $s_t^m$ is 2D data associated with the map, such as the terrain and distribution of resources, and $s_t^a$ is data associated with the agent, such as its position, health, and internal resources.

**Terrain:** Each map has a rock layer defining the height of each grid cell. Maps also include water, which is initially added to grid cells with a rock value below a preset sea-level. Throughout an episode, water flows downhill using a discrete flow simulation based on the height defined by this rock layer. In the experiments below, we use six global terrain configurations shown in Figure 2: **Ocean**, **Beach**, **Island**, **Lake**, **Isthmus**, and **Channel**. We conduct experiments at 64×64, 128×128, 256×256, 512×512, and 1024×1024 grid sizes. When growing or shrinking the grid size, we also scale the initial population size proportional to the area change of the overall grid. Once the simulation begins running, the population then fluctuates according to the dynamics of the environment.

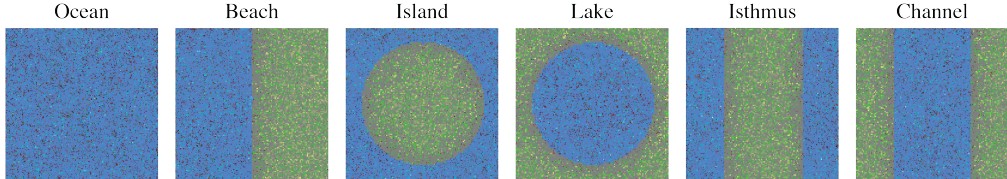

Figure 2: Overhead images of the terrains used in our experiments at 256×256 grid size. Small dark spots represent individual agents, while green and yellow represent concentrations of energy and biomass. These images were each captured 5000 steps into an episode after most biomass in the water has been eaten, but before the agents have evolved to survive on land for longer periods.

**Resources and Agent Health:** Agents must collect **water**, **energy**, and **biomass** to survive and reproduce in these environments. Initially, biomass and energy are distributed randomly throughout the environment but can be consumed and redistributed by individual agents.

Each action an agent takes expends a small amount of energy and water. The spent water returns to the landscape, where it flows downhill to the nearest body of water, while the energy is used up and destroyed. Biomass is used for reproduction; all agents require a fixed amount of biomass to exist and therefore cannot reproduce until they have collected enough excess biomass to donate to their offspring. Like water, the total biomass remains constant in the environment throughout the simulation, but unlike water, the biomass does not flow downhill or move on its own. Energy is not held constant but slowly grows on free-standing biomass. We designed these resource dynamics to provide agents with a slightly more challenging problem than collecting a single food source. For ablations on resource rules, see Appendix F.2.

Agents also have health points (HP) that they must maintain in order to avoid death. If an agent overspends a critical resource, their health will rapidly deteriorate, but they may use energy and water to slowly recover HP over time. Agents may also attack other agents, reducing the targets' HP. They also take a small but increasing amount of damage based on their age, meaning that agents eventually die of old age if not by other causes. When an agent's HP goes to zero and they die, whatever resources they were carrying are returned to the environment in the grid cell they were standing on. In this way, if an agent attacks and kills another agent, it may then consume their resources as a basic form of predation. For ablations on HP hyperparameters, see Appendix F.8.

**Sensors:** Agents sense the environment using four sensors. The **internal** sensor tells the agent about its age, health, and the resources it is carrying; the **external** sensor detects external resources in the same grid cell as the agent; a **compass**, inspired by the magnetoreceptors found in birds and other animals, detects the agent's global direction as a 4-way one-hot vector; and **vision** provides a $7 \times 7$ top-down rendered patch around the agent, containing three color channels as well as a fourth channel showing the local difference in elevation. For ablations on vision configurations, see Appendix F.6.

**Actions:** Agents have a finite set of discrete actions grouped into five categories. The first is **Rest**, which does nothing but costs the least energy. There are three **Move** actions: rotate clockwise, rotate counterclockwise, and move forward one grid cell. Multiple agents cannot occupy the same cell; if an agent tries to move into the same cell as another agent, or if two agents try to move to the same cell, their move action is canceled. The agents also have a single **Attack** action which kills all agents in the $3 \times 3$ square directly in front of them. For ablations on attack strength, see Appendix F.7. There are three separate **Eat** actions, one for each of the resource types. If an agent is in the same cell as a particular resource, taking the corresponding eat action will transfer a fixed amount of that resource from the environment to the agent's stomach, which has a fixed maximum capacity.

Finally, agents have a single **Reproduce** action that creates a new offspring via single-parent reproduction. This action is only successful if the agent has accumulated enough resources to donate to its new child. The newly created agent is created directly behind its parent. The child's policy contains a mutated copy of its parent's weights. The child also immediately inherits a fixed amount of its parent's resources.

In the experiments below, we often disable certain sensors or the attack action in order to test the ecological effects of agents with different abilities. We use three agent sensing combinations: internal+external resources only (**R**), internal+external+compass (**RC**), and internal+external+compass+vision (**RCV**). We use **+A** to note when attacking is allowed; for example (**RC+A**) indicates the agent has resource sensors, a compass, and can attack.

### 3.3 POLICIES

Our agent policies are implemented as small, memoryless MLPs. We opted for agents without memory in order to simplify the computational model, and reduce network parameters. The raw values for each sensor are flattened, normalized to the range $[-1, 1]$ and fed through separate linear layers with an output dimension of 64. These values are summed and fed through a 2-layer MLP with hidden dimension 64 and ReLU nonlinearities. The action space is discrete, so the final output is a linear layer with an output dimension equal to the number of possible actions. Actions are sampled

from a softmax distribution with a temperature that is allowed to evolve with the network weights. Weights are represented using half-precision bfloat16 values to support larger population sizes and faster run times. Depending on the number of sensors an agent is using, the network size for each agent is roughly between 10k-25k parameters. For ablations on network size and architecture, see Appendix F.5.

In all runs, the weights for the initial population of agents use Kaiming initialization (He et al., 2015); bias vectors are initialized to zero. Policies for agents born in the middle of an episode are created by copying their parent's weights and applying Gaussian noise with standard deviation $3 \times 10^{-2}$ to all mutable weight values. This mutation approach is similar to that used in other neuroevolution research (Such et al., 2017). While more complex approaches such as crossover are possible, we opt for this mechanism to simplify agent reproduction and study the effect of environmental scale in as simple a setting as possible. Empirically, we found that a larger mutation rate was necessary due to the use of low-precision floating-point numbers. We perform an ablation on mutation rates in Appendix F.4.

Agents also have a 3-channel trait controlling their color when rendered onto the map. The color trait mutates similarly to the network weights and determines what an agent looks like when observed by other agents with the vision sensor. In theory, this allows agents to distinguish between different types of other agents.

### 3.4 COMPUTATION AND REPRODUCIBILITY

All experiments are implemented in JAX (Bradbury et al., 2018) and run on Nvidia H100 and A40 GPUs using a state-of-the-art simulation environment. All experiments are run for 2M environment steps or until extinction unless otherwise noted, with per-seed wall-clock times ranging from under an hour for 64×64 grids to approximately 10 hours for 1024×1024 grids on one H100 GPU. Appendix E describes simulator architecture in more detail, reports additional scaling measurements, and discusses reproducibility considerations.

## 4 EXPERIMENTS

In the experiments below, we demonstrate the effects of different sensing modalities, environmental scale, and population size on the emergent behavior of agents.

### 4.1 LONG-DISTANCE RESOURCE GATHERING

In our first experiment, we study the emergent abilities of agents to travel long distances inland in order to find resources. We demonstrate that in environments with clear directional structure, agents with a compass (**RC**) will consistently develop much more effective long-range resource gathering behavior compared to a baseline agent class (**R**) without a compass. We also show that the emergence of this behavior is more consistent and effective at larger environmental scales.

We begin by discussing a visual example of the (**R**) agent class, shown in Figure 3. After 5000 time steps (a), all the initial agents on dry land have died due to lack of water. In the meantime, those in the water have rapidly increased in number due to the rich initial resources. At this early stage, the population does not yet contain sophisticated policies that can efficiently seek out resources. From this early peak in population, agents in the water occasionally stumble onto land (b). When they do, they often cannot find their way back to the ocean and die, depositing their biomass on land. By 50,000 time steps (c), this gradual process has nearly doubled the amount of uneaten biomass on land. This transfer of resources from water to land has caused the population to shrink because there is no longer enough biomass in the water to support a large population. By 125,000 steps (d), almost all biomass has been transferred to land, and the population is nearly extinct. The final agent dies around 185,000 steps (e).

This example demonstrates the ecological impact of the agents' inability to find their way back to water. Not only do the individuals die, but their deaths alter the distribution of resources in the environment in ways that have important consequences for the entire population. At the same time, this ecological catastrophe presents an opportunity for any agent that is able to successfully discover

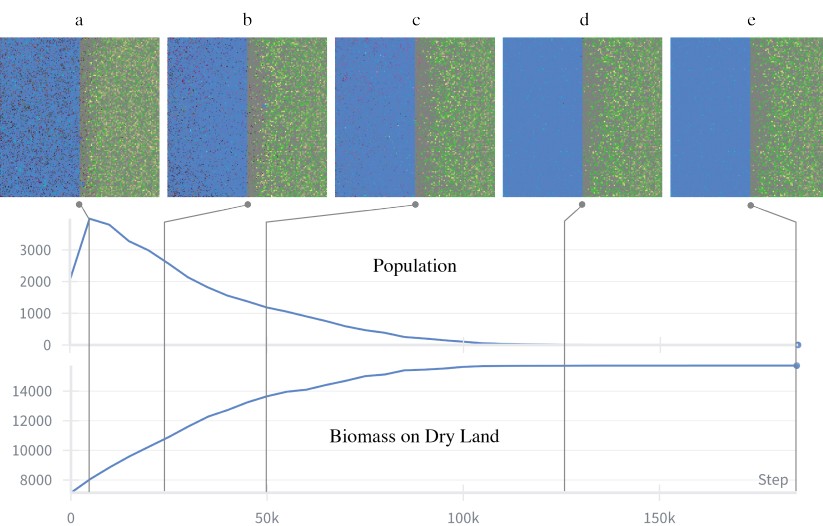

Figure 3: A representative trial with **(R)** agents in a $256 \times 256$ **Beach** environment. The top chart shows population over time, while the bottom chart shows the free biomass on dry land.

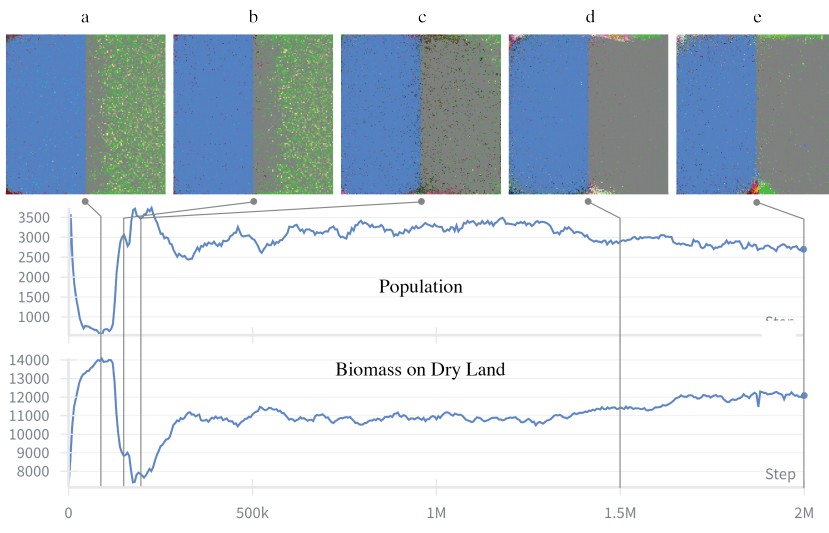

Figure 4: A representative trial with **(RC)** agents in a $256 \times 256$ **Beach** environment. The top chart shows population over time, while the bottom chart shows the free biomass on dry land.

a way to walk onto land, forage in the rich deposits of biomass left by their dead ancestors, and safely return to water.

In the next example, shown in Figure 4, we provide agents with an internal compass **(RC)**. At first, a similar pattern emerges (a) where the population shrinks, and the free biomass on land rapidly increases. However, in this example, some agents are able to use their compass to drift into the relative safety of the corners. At around 150,000 steps (b) some agents start to "mine" the biomass on land, using their compass to navigate back and forth from land to water. By 200,000 steps (c) these miners have cleared the entire beach. The population undergoes minor changes with agents forming clumps around the edges (d) but remains largely stable for the full 2,000,000 time steps (e).

We quantify these observations by running a full set of **Beach** experiments using grid sizes $1024 \times 1024$, $512 \times 512$, $256 \times 256$, $128 \times 128$, and $64 \times 64$, with initial populations of 32768, 8192, 2048, 512, and 128 agents respectively. We run each experiment with four random seeds and record

Table 1: Mining events in the **Beach** world in populations of agents with a compass **(RC)** compared to populations without a compass **(R)** across five world sizes and initial populations. ⛏ denotes the number of seeds that featured a mining event. 💀 denotes the number of seeds in which the population went extinct before 2M steps. See Table 3 in Appendix A for detailed data from each seed.

| | $1024 \times 1024$ | | $512 \times 512$ | | $256 \times 256$ | | $128 \times 128$ | | $64 \times 64$ | |
| | 32768 | | 8192 | | 2048 | | 512 | | 128 | |
| | ⛏ | 💀 | ⛏ | 💀 | ⛏ | 💀 | ⛏ | 💀 | ⛏ | 💀 |
| --- | --- | --- | --- | --- | --- | --- | --- | --- | --- | --- |
| **(RC)** | 4/4 | 0/4 | 4/4 | 0/4 | 4/4 | 0/4 | 3/4 | 1/4 | 2/4 | 3/4 |
| **(R)** | 1/4 | 0/4 | 0/4 | 4/4 | 0/4 | 4/4 | 0/4 | 4/4 | 0/4 | 4/4 |

whether or not mining behavior emerges, which we report as a drop of at least 10% in the free biomass on dry land from the historical high point. We also record the onset time of the mining event, determined by the local maximum in biomass on land, and the percentage drop in biomass. A summary of mining and extinction events can be found in Table 1 with full details on each seed in Table 3 in Appendix A. Mining behavior consistently emerges in **(RC)** agents compared to **(R)** agents. Mining emerges more consistently in larger environments and occurs in all seeds starting at the 256×256 scale. At smaller scales, populations become less stable with a higher likelihood of going extinct earlier on.

To explore these effects more closely, we run additional seeds with **(R)** and **(RC)** agents at 512×512 world size (32 seeds) and 128×128 world size (64 seeds) and use a Kaplan-Meier estimator to measure the probability of extinction and mining as a function of simulator time. We find that extinction is more rare and takes longer to occur at larger world sizes, while mining is more frequent and happens sooner at smaller grid sizes. We also fit a Cox proportional hazards model to compare the timing of mining discovery and timing of extinction across world sizes and compass usage, similarly finding that a larger world size substantially reduces the rate of extinction and enables much more rapid mining discovery than in smaller worlds. Enabling compass produces a similar effect. Full details can be found in Appendix B.

Table 2: Mining and extinction events in the **Island**, **Lake**, **Isthmus**, and **Channel** terrains in populations of agents with a compass **(RC)** compared to those without a compass **(R)** at the $512 \times 512$ grid size. ⛏ denotes the number of seeds that featured a mining event. 💀 denotes the number of seeds in which the population went extinct before 2M steps. See Table 4 in Appendix A for detailed data from each seed.

| | Island | | Lake | | Isthmus | | Channel | |
| | ⛏ | 💀 | ⛏ | 💀 | ⛏ | 💀 | ⛏ | 💀 |
| --- | --- | --- | --- | --- | --- | --- | --- | --- |
| **(RC)** | 4/4 | 0/4 | 2/4 | 2/4 | 4/4 | 0/4 | 3/4 | 1/4 |
| **(R)** | 3/4 | 1/4 | 2/4 | 4/4 | 1/4 | 3/4 | 0/4 | 4/4 |

We run similar experiments in the **Island**, **Lake**, **Isthmus**, and **Channel** terrains at the $512 \times 512$ grid size with **(R)** and **(RC)** agents in order to investigate the impact of the terrain configuration on the emergence of mining behavior. Mining and extinction events for these experiments can be found in Table 2, while detailed information about each run can be found in Table 4 in Appendix A. As before, equipping agents with a compass yields earlier and more consistent emergence of mining behavior, as well as more stable populations with lower extinction rates. **(RC)** agents adapt more easily to the island and isthmus environments than to the lake and channel, where they occasionally go extinct. We hypothesize that this is because walking either straight east or west will always take an agent back to water in the island and isthmus, but there is no easy rule like this to follow for the channel and lake. The **(R)** agent cannot reliably adapt to any of these environments, but struggles least on the island. See Figure 12 in Appendix D for a visual of mining in the island terrain.

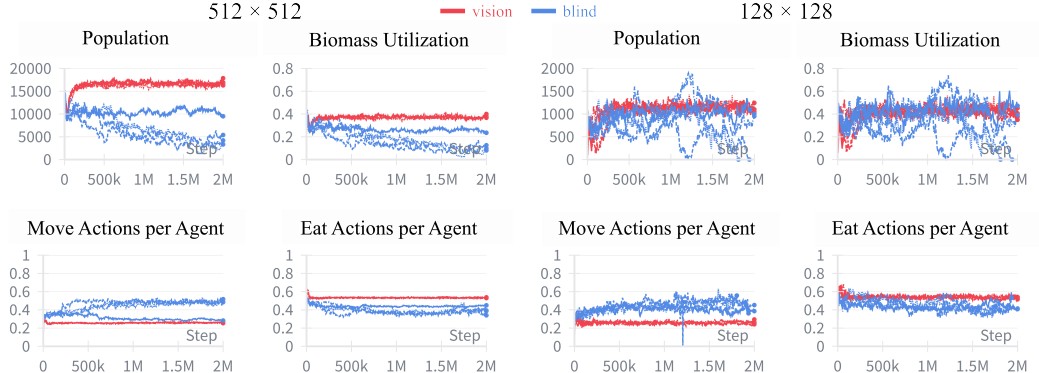

Figure 5: Population, biomass utilization, move actions per agent, and eat actions per agent in 512×512 and 128×128 **Ocean** worlds. Biomass utilization, move actions per agent, and eat actions per agent are averaged across agents at each time step. Each plot shows four seeds with vision agents **(RCV + A)** (red) and four seeds with blind agents **(RC + A)** (blue). With vision, agent populations are larger and more stable, with greater biomass utilization as a result of less moving and more frequent and effective eating. At larger scales, these dynamics are more stable with less variance across runs, suggesting that in larger environments, the vision sensor offers an increasing advantage over blindness for adapting foraging behavior. See Figure 9 in Appendix C for a comparison of grid sizes 1024×1024 and 256×256.

## 4.2 VISUAL FORAGING AND PREDATION

We also conduct experiments in the **Ocean** environment to determine whether agents can adapt to use vision sensors to effectively locate free resources or prey on other agents. We compare the performance of **(RCV+A)** agents with resource sensing, a compass, and a $7 \times 7$ visual sensor against **(RC+A)** agents with only resource and compass sensors.

Figure 5 plots population, biomass utilization, movement actions, and eat actions for agents in $512 \times 512$ and $128 \times 128$ grid sizes. Plots for other grid sizes can be found in Figure 9 in Appendix C, and summary statistics of these plots can be found in Table 5 in Appendix A. Biomass utilization is computed as the percentage of the total biomass in the environment that is currently inside the agents. We find that in episodes where agents have vision sensors (red lines), they spend less time moving and more time eating. Additionally, these runs support larger populations and have greater biomass utilization. These dynamics indicate that the agents have adapted to use their visual sensing abilities to forage for food more efficiently. When comparing across environmental scales, we find that the variance between runs decreases substantially in larger environments. Unexpectedly, runs that include vision sensors have much more stable dynamics, with less variation in these metrics over time.

Figure 6 shows the averaged attacks per agent and homicides per attack for these runs. These statistics show that agents with vision attack relatively infrequently, but maintain a successful kill rate of approximately 75%. In large grids, agents without vision sometimes also have high kill rates, though with much more unstable dynamics. Unfortunately, there are confounding factors that make it difficult to determine the underlying causes of this observation. Through visual inspection (see Figure 13 in Appendix D), we have found that sometimes these high kill rates occur when clusters of agents build up in one location, and that these clusters almost never form in runs with visual agents. This may also indicate that agents with vision can adapt to avoid each other. In smaller environments, the difference between visual agents and blind agents is less pronounced, but populations of visual agents remain far more stable.

## 4.3 ABLATIONS

In Appendix F, we perform ablations on specific hyperparameters such as initial population size (F.1), resource rules (F.2), mutation rate (F.4), agent network configuration (F.5), vision range (F.6), attack strength (F.7), and agent HP (F.8) to measure the sensitivity of emergent behaviors to simu-

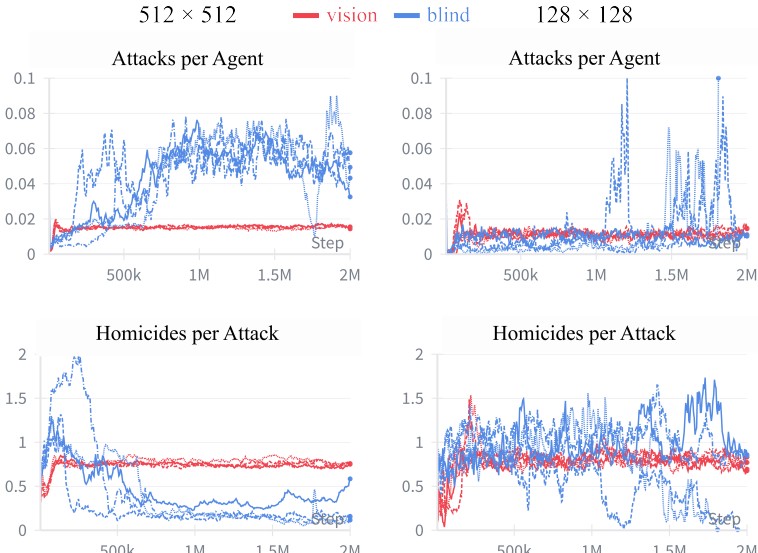

Figure 6: Attacks per agent and homicides per attack, averaged across agents, in 512×512 and 128×128 **Ocean** worlds. Each plot shows four seeds with vision agents **(RCV + A)** (red) and four seeds with blind agents **(RC + A)** (blue). With vision, agents attack less frequently but with much higher precision, suggesting the emergence of specialized predation behaviors. This dynamic is more pronounced at larger scales. See Figure 10 in Appendix C for a comparison of grid sizes 1024×1024 and 256×256.

lator configuration. Additionally, we perform experiments in Appendix F.3 to verify that dynamics observed in larger environments are not simply due to self-averaging.

## 5 CONCLUSION

Our study highlights the promise of open-ended ecological settings as a powerful paradigm for investigating the emergence of intelligence. Through experiments in worlds of varying scale and complexity, we find that both environmental scale and sensory richness greatly contribute to the development of sophisticated sensing and decision-making policies. Taken together, these results suggest that ecological and evolutionary pressures, when coupled with rich environments, may provide a natural pathway for the development of increasingly intelligent artificial agents. Furthermore, our experiments demonstrate that rich behaviors can be evolved in large populations within a reasonable hardware budget without resorting to explicit objectives. We hope that these results will enable further exploration of ecology as a mechanism for developing and understanding artificial intelligence.

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

# A    ADDITIONAL DATA

Table 3: Mining events in the **Beach** world in populations of agents with a compass **(RC)** compared to populations without a compass **(R)** across five scales. ⛏ denotes whether a run had a mining event. 📉 denotes the percent decrease in free biomass on dry land during the mining event, and ⏱ denotes the approximate timestep at which free dry biomass reached the local maximum indicating the start of the mining event. For runs with multiple mining events, we report the first one. 💀 denotes the timestep at which the run went extinct, or n/a if it lasted for 2M steps.

| | 1024x1024 32768 | | | | 512x512 8192 | | | | 256x256 2048 | | | | 128x128 512 | | | | 64x64 128 | | | |
|---|---|---|---|---|---|---|---|---|---|---|---|---|---|---|---|---|---|---|---|---|
| | **Compass On (RC)** | | | | | | | | | | | | | | | | | | | |
| Seed | ⛏ | ⏱ | 📉 | 💀 | ⛏ | ⏱ | 📉 | 💀 | ⛏ | ⏱ | 📉 | 💀 | ⛏ | ⏱ | 📉 | 💀 | ⛏ | ⏱ | 📉 | 💀 |
| 0 | ✓ | 35k | 11.1 | n/a | ✓ | 105k | 33. | n/a | ✓ | 145k | 49.3 | n/a | ✓ | 105k | 72.5 | n/a | x | | | 55 |
| 1 | ✓ | 90k | 16.2 | n/a | ✓ | 50k | 17.2 | n/a | ✓ | 50k | 46.6 | n/a | x | | | 140 | ✓ | 70k | 85.3 | 675 |
| 2 | ✓ | 40k | 15. | n/a | ✓ | 30k | 14.3 | n/a | ✓ | 55k | 34.7 | n/a | ✓ | 50k | 50.7 | n/a | ✓ | 10k | 78. | n/a |
| 3 | ✓ | 60k | 15. | n/a | ✓ | 45k | 15.6 | n/a | ✓ | 90k | 47.4 | n/a | ✓ | 20k | 52. | n/a | x | | | 135 |
| Avg | 4/4 | 34k | 14.3 | 0/4 | 4/4 | 58k | 20. | 0/4 | 4/4 | 85k | 44.5 | 0/4 | 3/4 | 58k | 58.4 | 1/4 | 2/4 | 40k | 81.7 | 3/4 |
| | **Compass Off (R)** | | | | | | | | | | | | | | | | | | | |
| Seed | ⛏ | ⏱ | 📉 | 💀 | ⛏ | ⏱ | 📉 | 💀 | ⛏ | ⏱ | 📉 | 💀 | ⛏ | ⏱ | 📉 | 💀 | ⛏ | ⏱ | 📉 | 💀 |
| 0 | x | | | n/a | x | | | 945k | x | | | 185k | x | | | 65k | x | | | 45k |
| 1 | ✓ | 275k | 12.4 | n/a | x | | | 1085k | x | | | 110k | x | | | 55k | x | | | 40 |
| 2 | x | | | n/a | x | | | 1070k | x | | | 165k | x | | | 55k | x | | | 25 |
| 3 | x | | | n/a | x | | | 280k | x | | | 125k | x | | | 90k | x | | | 40 |
| Avg | 1/4 | 275k | 12.4 | 0/4 | 0/4 | | | n/a | 0/4 | | | 4/4 | 0/4 | | | 4/4 | 0/4 | | | 4/4 |

Table 4: Mining events in the **Island**, **Lake**, **Isthmus**, and **Channel** among **(RC)** agents compared to **(R)** agents. Grid size is 512×512 with an initial population of 8192.

| | Island | | | | Lake | | | | Isthmus | | | | Channel | | | |
|---|---|---|---|---|---|---|---|---|---|---|---|---|---|---|---|---|
| | Compass On (RC) | | | | | | | | | | | | | | | |
| Seed | ⛏ | 😵 | 〽 | 💀 | ⛏ | 😵 | 〽 | 💀 | ⛏ | 😵 | 〽 | 💀 | ⛏ | 😵 | 〽 | 💀 |
| 0 | ✓ | 40k | 94.3 | n/a | x | | | 45k | ✓ | 45k | 33. | n/a | ✓ | 50k | 27.4 | n/a |
| 1 | ✓ | 30k | 94.3 | n/a | ✓ | 30k | 22.5 | n/a | ✓ | 55k | 37.3 | n/a | ✓ | 80k | 30.8 | n/a |
| 2 | ✓ | 30k | 94.8 | n/a | x | | | 45k | ✓ | 40k | 28.1 | n/a | ✓ | 50k | 41. | n/a |
| 3 | ✓ | 70k | 95.1 | n/a | ✓ | 35k | 25.3 | n/a | ✓ | 30k | 41.5 | n/a | x | | | 80k |
| Avg | 4/4 | 43k | 94.6 | 0/4 | 2/4 | 33k | 23.9 | 2/4 | 4/4 | 43k | 35. | 0/4 | 3/4 | 60k | 33.1 | 1/4 |
| | Compass Off (R) | | | | | | | | | | | | | | | |
| Seed | ⛏ | 😵 | 〽 | 💀 | ⛏ | 😵 | 〽 | 💀 | ⛏ | 😵 | 〽 | 💀 | ⛏ | 😵 | 〽 | 💀 |
| 0 | x | | | 185k | x | | | 125k | x | | | 145k | x | | | 100k |
| 1 | ✓ | 75k | 30.9 | n/a | x | | | 130k | x | | | 260k | x | | | 105k |
| 2 | ✓ | 85k | 47. | n/a | ✓ | 75k | 23.4 | 1400k | ✓ | 95k | 18.8 | n/a | x | | | 105k |
| 3 | ✓ | 115k | 39.5 | n/a | ✓ | 70k | 24.3 | 1250k | x | | | 140k | x | | | 100k |
| Avg | 3/4 | 92k | 39.1 | 1/4 | 2/4 | 73k | 23.9 | 4/4 | 1/4 | 95k | 18.8 | 3/4 | 0/4 | | | 4/4 |

Table 5: Population and foraging data for Vision and Blind agents in the **Ocean** world at different scales. 👥 denotes population size. ⚙ denotes biomass utilization, the percentage of total biomass in the system that is currently inside the agents' stomachs. 🏃 denotes move actions per agent, the fraction of time an agent took a move action, averaged across agents. Similarly, 😊 denotes the proportion of agents' actions that took an eat action. Reported values of 👥, ⚙, 🏃, and 😊 are averages of the metric over all steps for the seed.

| | 1024x1024 32768 | | | | 512x512 8192 | | | | 256x256 2048 | | | | 128x128 512 | | | |
|---|---|---|---|---|---|---|---|---|---|---|---|---|---|---|---|---|
| | Vision (RCV+A) | | | | | | | | | | | | | | | |
| Seed | 👥 | ⚙ | 🏃 | 😊 | 👥 | ⚙ | 🏃 | 😊 | 👥 | ⚙ | 🏃 | 😊 | 👥 | ⚙ | 🏃 | 😊 |
| 0 | 58354 | .396 | .402 | .426 | 16413 | .369 | .257 | .532 | 4195 | .382 | .255 | .536 | 1126 | .426 | .261 | .541 |
| 1 | 65107 | .365 | .257 | .529 | 16200 | .364 | .256 | .533 | 4142 | .377 | .258 | .536 | 1103 | .415 | .262 | .537 |
| 2 | 64735 | .363 | .257 | .530 | 16560 | .372 | .257 | .533 | 4269 | .388 | .260 | .536 | 1041 | .392 | .254 | .542 |
| 3 | 64595 | .362 | .256 | .532 | 16206 | .365 | .255 | .531 | 4249 | .388 | .259 | .533 | 1100 | .414 | .257 | .538 |
| Avg | 63198 | .371 | .293 | .504 | 16345 | .368 | .256 | .532 | 4213 | .384 | .258 | .535 | 1092 | .412 | .259 | .539 |
| | Blind (RC+A) | | | | | | | | | | | | | | | |
| Seed | 👥 | ⚙ | 🏃 | 😊 | 👥 | ⚙ | 🏃 | 😊 | 👥 | ⚙ | 🏃 | 😊 | 👥 | ⚙ | 🏃 | 😊 |
| 0 | 22734 | .143 | .446 | .372 | 6549 | .173 | .442 | .400 | 2036 | .209 | .433 | .461 | 545 | .219 | .465 | .427 |
| 1 | 23670 | .151 | .426 | .388 | 6939 | .180 | .438 | .396 | 3107 | .321 | .390 | .416 | 1082 | .444 | .416 | .460 |
| 2 | 24745 | .159 | .358 | .426 | 5448 | .136 | .474 | .374 | 1587 | .163 | .440 | .433 | 979 | .421 | .404 | .436 |
| 3 | 20004 | .130 | .469 | .359 | 10354 | .265 | .311 | .439 | 3164 | .324 | .321 | .450 | 788 | .317 | .431 | .477 |
| Avg | 22788 | .146 | .425 | .386 | 7323 | .189 | .416 | .402 | 2473 | .254 | .396 | .440 | 848 | .351 | .429 | .450 |

Table 6: Predation data for Vision and Blind agents in the **Ocean** world at different scales. 🗡 denotes attacks per agent, the fraction of time an agent took the attack action, averaged across all agents. The 🎯 denotes homicides per attack, the number of agents killed per attack action. Reported values of 🗡 and 🎯 are averages over all steps for the seed. Note the outlier of 0.562 for 🗡 of seed 3 in the 128×128 (**RC+A**) run. This seed went extinct before 2M steps, resulting in a spike in the attacks per agent metric due to dwindling population size.

| | 1024x1024 32768 | | 512x512 8192 | | 256x256 2048 | | 128x128 512 | |
|---|---|---|---|---|---|---|---|---|
| | Vision (RCV+A) | | | | | | | |
| Seed | 🗡 | 🎯 | 🗡 | 🎯 | 🗡 | 🎯 | 🗡 | 🎯 |
| 0 | .011 | .776 | .015 | .731 | .013 | .784 | .010 | .788 |
| 1 | .016 | .733 | .015 | .741 | .014 | .780 | .011 | .793 |
| 2 | .016 | .707 | .015 | .737 | .014 | .750 | .012 | .771 |
| 3 | .016 | .712 | .015 | .780 | .013 | .759 | .011 | .765 |
| Avg | .015 | .732 | .015 | .747 | .014 | .758 | .011 | .779 |
| | Blind (RC+A) | | | | | | | |
| Seed | 🗡 | 🎯 | 🗡 | 🎯 | 🗡 | 🎯 | 🗡 | 🎯 |
| 0 | .070 | .218 | .042 | .437 | .005 | 1.074 | .016 | .558 |
| 1 | .054 | .300 | .044 | .383 | .016 | .871 | .006 | 1.032 |
| 2 | .076 | .236 | .048 | .238 | .031 | .383 | .010 | .979 |
| 3 | .059 | .203 | .044 | .456 | .033 | .446 | .562 | .841 |
| Avg | .065 | .239 | .045 | .378 | .021 | .693 | .148 | .853 |

## B  DEMOGRAPHY

We compare rates of extinction and mining emergence dependent on world size and agent access to global direction via the compass sensor. Extinction is defined as the population reaching zero individuals, while mining behavior is defined as it was in earlier experiments (a 10% drop in free biomass on land).

Runs are grouped based on both world size and compass access. We simulated 64 runs each of $128 \times 128$ with no compass and $128 \times 128$ with compass, and 32 runs each of $512 \times 512$ with no compass and $512 \times 512$ with compass. For each of the four groups, we fit a non-parametric Kaplan–Meier estimator to the time-to-extinction distribution (treating runs that did not go extinct within 2M steps as right-censored). In Figures 7 and 8, we observe that worlds tend to survive longer and produce mining more quickly if they are large or if agents have compass information.

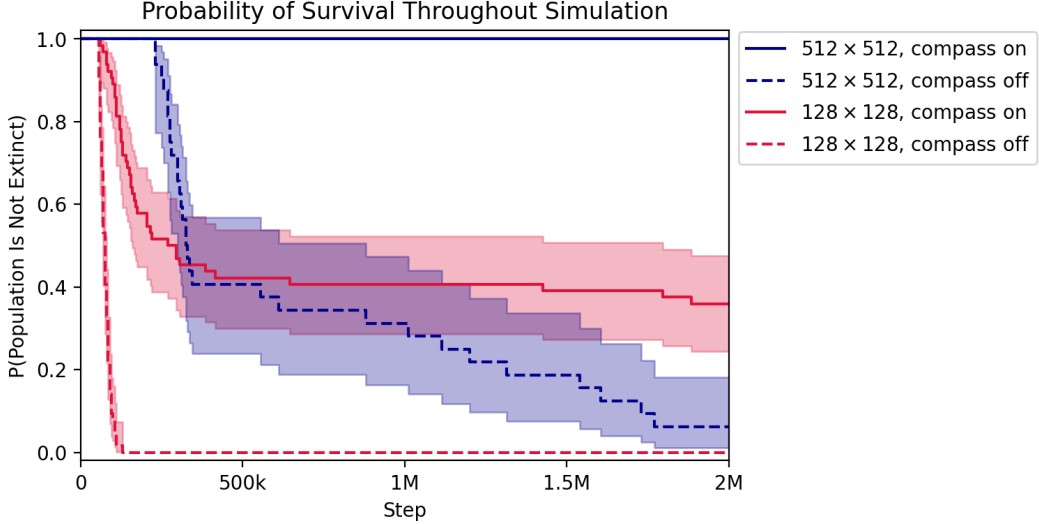

Figure 7: Probability the population has not gone extinct given a world size, compass access, and timestep. We fit a unique non-parametric Kaplan-Meier estimator for each group of runs (e.g. $512 \times 512$, compass on). The shaded areas represent the 95% confidence intervals for each curve. Each $512 \times 512$ group fits the estimator on 32 runs; each $128 \times 128$ group fits on 64 runs.

We compare emergence and extinction timing with a Cox proportional hazards model, a semi-parametric survival model relating covariates to event hazard rates (Table 7). Runs that did not experience an emergence or extinction event by the end of 2M time steps were treated as censored. For extinction, both larger world size and compass use substantially reduce the hazard; the extinction rate in $128 \times 128$ is about 25 times greater than the corresponding rate in $512 \times 512$ worlds ($HR=0.04$). Enabling the compass produces an effect of a similar magnitude ($HR=0.03$). For mining emergence, larger worlds generate mining behavior much more rapidly ($HR=6.07$), and compass use accelerates emergence dramatically ($HR=34.59$). The models show good discriminative ability between groups (concordance 0.80 for extinction, 0.85 for mining). We find that larger worlds can sustain larger populations, reducing the chance of hitting the absorbing zero boundary while simultaneously exploring a larger phenotype space due to greater quantities of mutation as compared to smaller worlds.

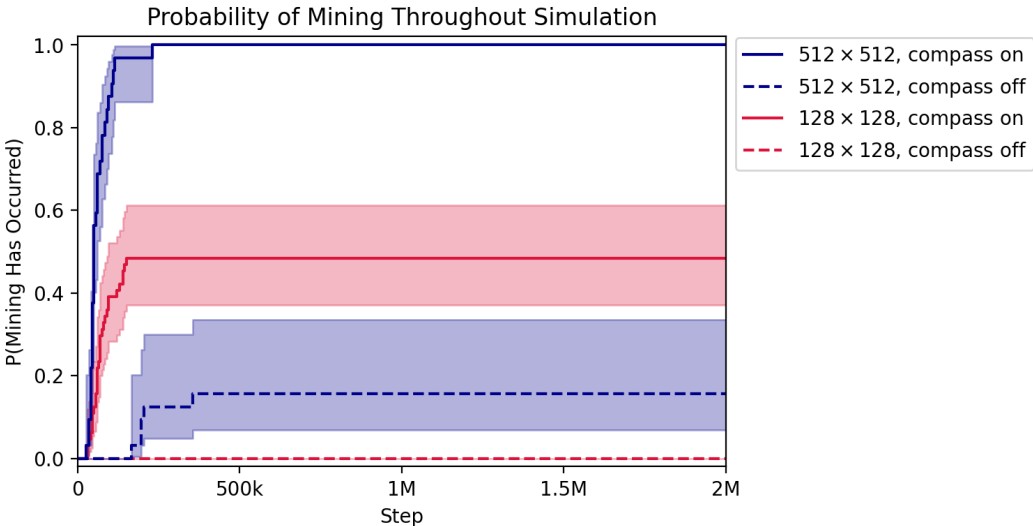

Figure 8: Probability mining has evolved within the population given a world size, compass access, and timestep. We fit a unique non-parametric Kaplan-Meier estimator for each group of runs (e.g. $512 \times 512$, compass on). The shaded areas represent the $95\%$ confidence intervals for each curve. Each $512 \times 512$ group fits the estimator on 32 runs; each $128 \times 128$ group fits on 64 runs.

Table 7: Cox proportional hazards models for extinction and emergence events with a $128 \times 128$, compass off baseline. Hazard ratios (HR) less than 1 indicate reduced hazard (longer time to event); for example, extinction $HR = 0.04$ for $512 \times 512$ means extinction occurs at $0.04$ times the rate of $128 \times 128$. There are 192 total runs: 32 seeds with compass on and 32 seeds with compass off in the $512 \times 512$ world, and 64 seeds with compass on and 64 seeds with compass off in the $128 \times 128$ world.

| | Extinction | | | Mining | | |
|---|---|---|---|---|---|---|
| **Covariate** | HR | 95% CI | $p$-value | HR | 95% CI | $p$-value |
| Size $512 \times 512$ | 0.04 | 0.02–0.07 | $< 0.005$ | 6.07 | 3.61–10.20 | $< 0.005$ |
| Compass On | 0.03 | 0.01–0.05 | $< 0.005$ | 34.59 | 13.51–88.57 | $< 0.005$ |
| **Concordance** | 0.80 | | | 0.85 | | |
| **Events / Total** | 135 / 192 | | | 68 / 192 | | |

## C   ADDITIONAL FORAGING AND PREDATION RESULTS

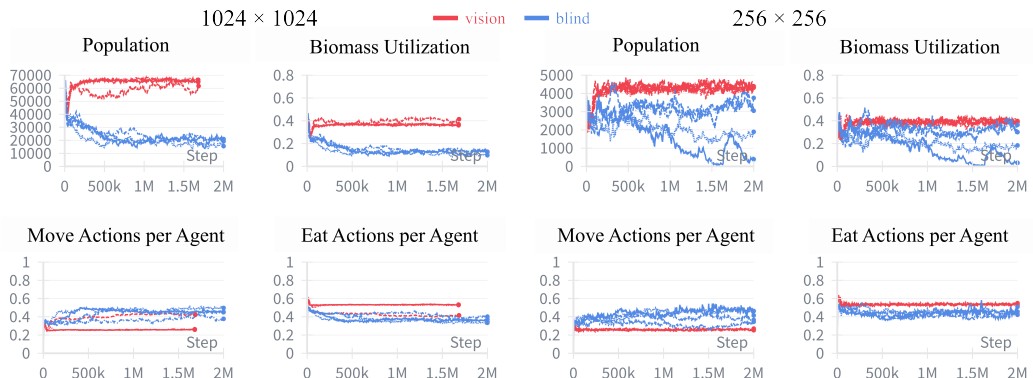

Figure 9: Foraging behaviors in the **Ocean** world at different scales. Enabling vision in agents (red) results in more stable populations and greater biomass utilization as a result of more frequent and effective eating and less movement. At larger scales, these dynamics are more stable with less variance across runs. Note that due to limitations on GPU hours, the vision seeds at the 1024×1024 scale ran for 1.7M instead of 2M steps.

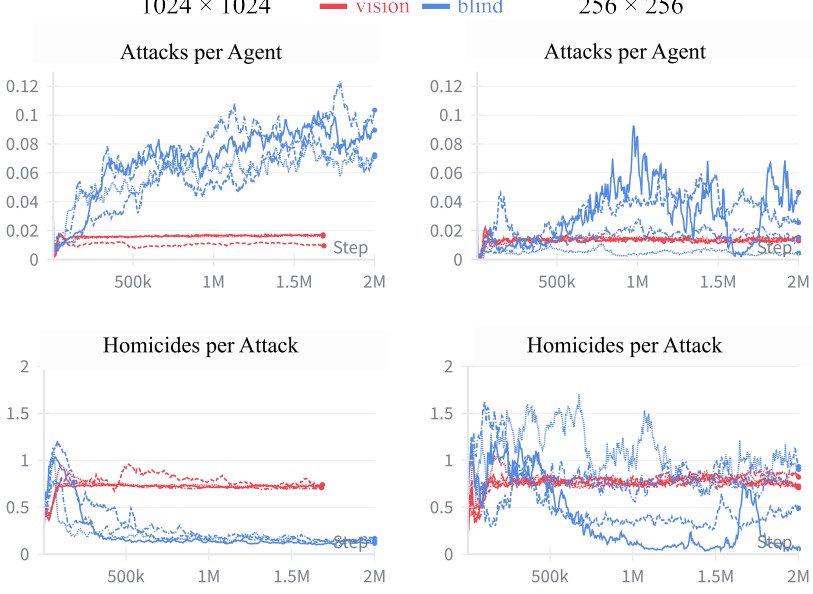

Figure 10: Predation behaviors in the **Ocean** world at different scales. With vision (red), agents attack less frequently but with much higher precision, suggesting the emergence of specialized predation behaviors in addition to foraging capabilities (Figure 9). This dynamic is again more pronounced at larger scales, with more variance in the behaviors of blind agents (blue) in smaller world sizes.

## D   ADDITIONAL EXPERIMENT VISUALIZATIONS

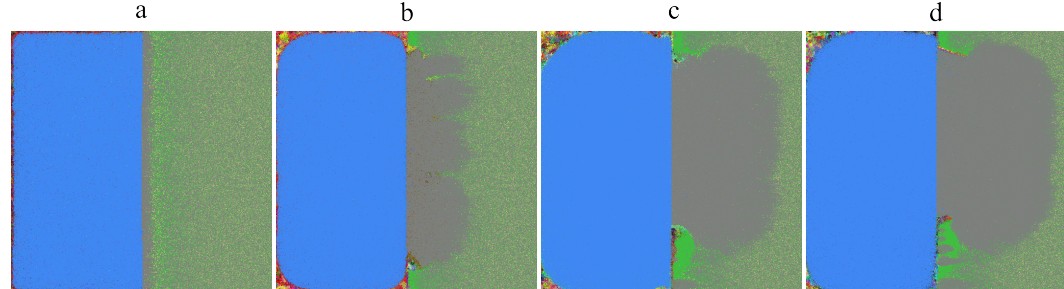

Figure 11: Larger scale environments induce more varied and complex ecologies that give rise to unique visual patterns. In this 1024×1024 **Beach** world, **(RC)** agents initially died at the shore (a), transferring biomass to dry land. They then developed mining behavior, clearing biomass from the beach (b). When biomass re-accumulated on the ends of the beach (c), subpopulations of agents re-developed mining behavior, traveling inland on these mini biomass beaches to extract the resources (d).

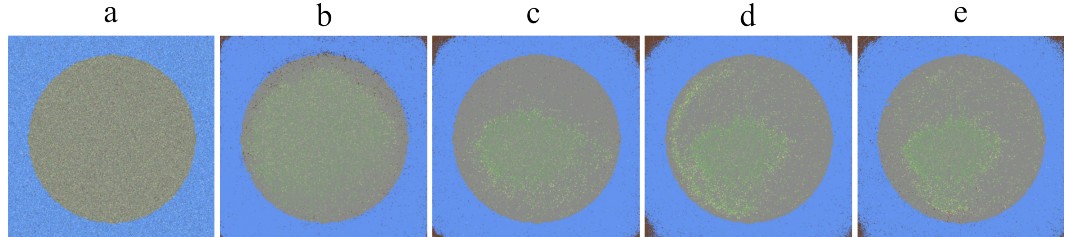

Figure 12: Mining behavior emerges in varied terrains. In this 512×512 **Island** world with **(RC)** agents, different regions of the shore are mined throughout the run (b) (c). Biomass is transferred back to parts of the island once mining becomes difficult and agents die before they can return to water (d). Agents adapt to re-mine this wall of accumulated biomass, clearing it from shore (e).

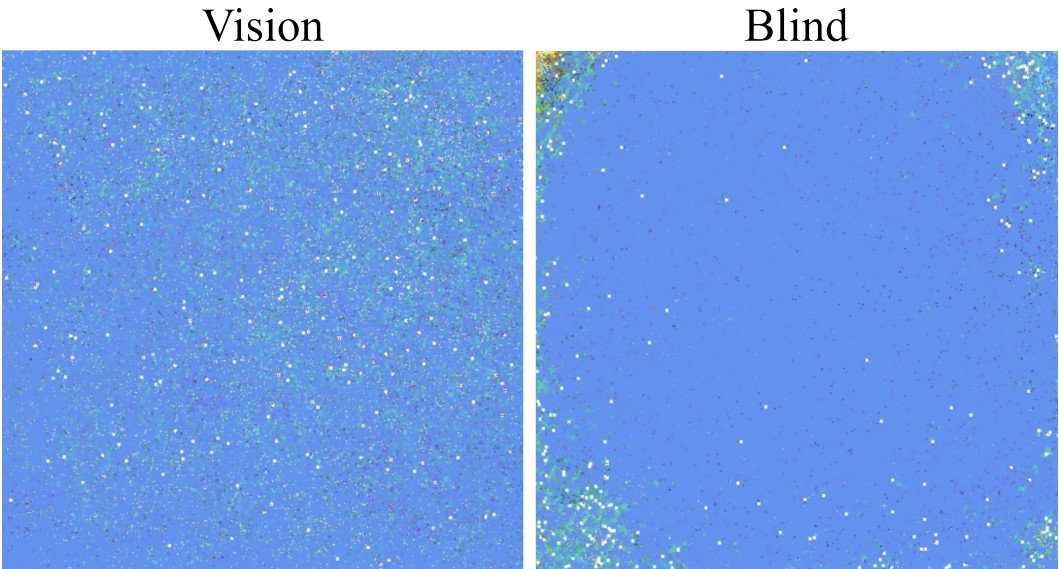

Figure 13: Representative runs in a 512×512 **Ocean** world where agents have attacking enabled: vision **(RCV+A)** agents on the left and blind **(RC+A)** agents on the right. The white squares indicate an agent is attacking at that location. With vision, agents adapt to locate free resources throughout the ocean world and prey on other agents more effectively (see Figures 5 and 6). Blind agents explore the ocean for resources to a much lesser extent, instead adapting policies for clustering in the corners of the world.

# E   DETAILS ON COMPUTATION AND REPRODUCIBILITY

We report details on simulation design, runtime, and experiment reproducibility.

**Simulator design.** We conduct all experiments using a JAX-based, GPU-accelerated simulator designed for large-scale multi-agent ecosystems with open-ended populations. Our simulator computes every part of the environment step on the GPU, including terrain dynamics, resource flows, agent updates, and measurement, so that policy evaluation and simulation share a single JAX compilation pipeline. The simulator exposes a minimal population-level interface: given current map state, agent states, and heritable traits, a population algorithm supplies actions and updated traits and handles births, deaths, and environmental transitions. This separation allows the same environment implementation to be reused with different evolutionary algorithms, reinforcement learning methods, or other population update rules. All operations are implemented as pure JAX functions combined with vectorization and just-in-time compilation, enabling fusion into a small number of GPU kernels. This yields the throughput required for our scaling experiments, with runtime measurements for a single H100 GPU given in Table 8.

A key feature of the simulator is its built-in tooling, including a measurement API for computing and recording aggregate statistics such as biomass utilization, movement and eating rates, attack and homicide rates, and extensive environment-level parameters at each step, which we use directly in our figures and tables. In addition, an interactive 3D viewer renders terrain, water, resources, and agents from generated reports, and supports inspection of individual agents (Figure 1). These tools were critical for discovering and validating qualitative phenomena such as long-range resource mining and predation patterns that are not obvious from scalar metrics alone.

From a high-performance computing perspective, our simulator fills a gap relative to existing simulators. General-purpose agent-based frameworks such as MASON (Luke et al., 2005), FLAME-GPU (Richmond et al., 2024), or Griddly (Bamford et al., 2020) offer high configurability but require substantial environment engineering and do not directly target the open-ended, evolving populations at the scales we study. Conversely, GPU-native multi-agent RL environments emphasize fixed numbers of agents and task-specific reward structures. We combine GPU-native execution, explicit support for births and deaths with mutable traits, and configurable resource and terrain dynamics, enabling open-ended populations of over $10^5$ agents while remaining interoperable with diverse population algorithms. This combination yields a state-of-the-art platform for scaling ecological experiments like those in this paper and a useful foundation for future work on emergent behavior in large populations. Our simulator is open-sourced at **url withheld to preserve anonymity.**

**Runtime data.** Table 8 shows runtime data across different grid sizes and maximum populations of agents. Each run was conducted on a single H100 GPU for a maximum of 2M steps. Note that while this table shows run-times for **(RC)** and **(RCV + A)** agents with 2 MLP layers and 64 channels per layer ($\sim$ 22k parameters total), the runtime varies with the size of the agent networks, the available sensors, and the level of logging.

Our largest runs at the $1024 \times 1024$ grid size take $\sim$10 hours in the **Beach** terrain and $\sim$14 hours in the **Ocean** terrain on one H100, with memory usage of roughly 60GB. **Ocean** runs are slightly slower due to the enabled agent attack action and vision sensing.

Table 8: Runtime data in the **Beach** and **Ocean** worlds across different grid sizes and maximum populations of agents. The **(RC)** agents each have a 22,465 parameter neural network. The **(RCV + A)** agents each have a 22,529 parameter neural network. Simulation steps per second (hz) and total extrapolated wall-clock runtime in hours for 2M steps are reported. Effective runtime is less for seeds in which populations go extinct before 2M steps.

| | $1024 \times 1024$ | | $512 \times 512$ | | $256 \times 256$ | | $128 \times 128$ | | $64 \times 64$ | |
| | 131072 | | 32768 | | 16384 | | 8192 | | 4096 | |
| Environment | hz | runtime | hz | runtime | hz | runtime | hz | runtime | hz | runtime |
|---|---|---|---|---|---|---|---|---|---|---|
| Beach (RC) | 55 | 10.1 hrs | 190 | 2.9 hrs | 315 | 1.8 hrs | 535 | 1.0 hrs | 750 | 0.7 hrs |
| Ocean (RCV + A) | 40 | 13.9 hrs | 142 | 3.9 hrs | 235 | 2.4 hrs | 415 | 1.3 hrs | 750 | 0.7 hrs |

**Reproducibility.** Due to the large and randomized nature of our environments, there can be substantial differences in the precise outcomes of each experiment from run to run. As a result, we report the results of four different random seeds for each experiment. Unfortunately, due to nondeterminism in some low-level GPU operations, we find that repeating an experiment with the same random seed does not reliably produce the same results. In particular, modern GPUs may execute parallel floating-point operations such as summations in different orders across runs and fuse operations in ways that change rounding behavior. Very small differences arising from finite-precision arithmetic can accumulate over many steps into noticeable drift in the simulation trajectory. The use of 16-bit weights can coarsen these rounding effects, but is not the primary source. Despite differences across runs, clear high-level patterns emerge when the results of multiple seeds are taken in aggregate.

# F    ABLATIONS

We present a series of ablations investigating the effects of specific hyperparameters and robustness of results under different environment and agent configurations.

## F.1    INITIAL POPULATION SIZE SWEEP FOR FIXED MAP SIZE

We evaluate the effect of initial population size in a $256 \times 256$ **Beach** world on the consistency of mining behavior and population stability. Results reported are for **(RC)** agent populations. Agents without compass **(R)** had 0% mining rate and 100% extinction rate across all initial population sizes except 8192, the highest tested, with 25% mining and 75% extinction frequency. Note that mining and extinction frequencies need not add to 100%: it can be the case that a population develops mining but then goes extinct, for instance, or that a population lasts for the full 2M steps of our experiments without developing the mining behavior.

In Figure 14, we find that the initial population size has no significant effect on mining consistency or extinction rates: populations starting with only 128 agents evolve mining behaviors as frequently as populations starting with 8192 agents, and extinction rates are largely stable across initial population size.

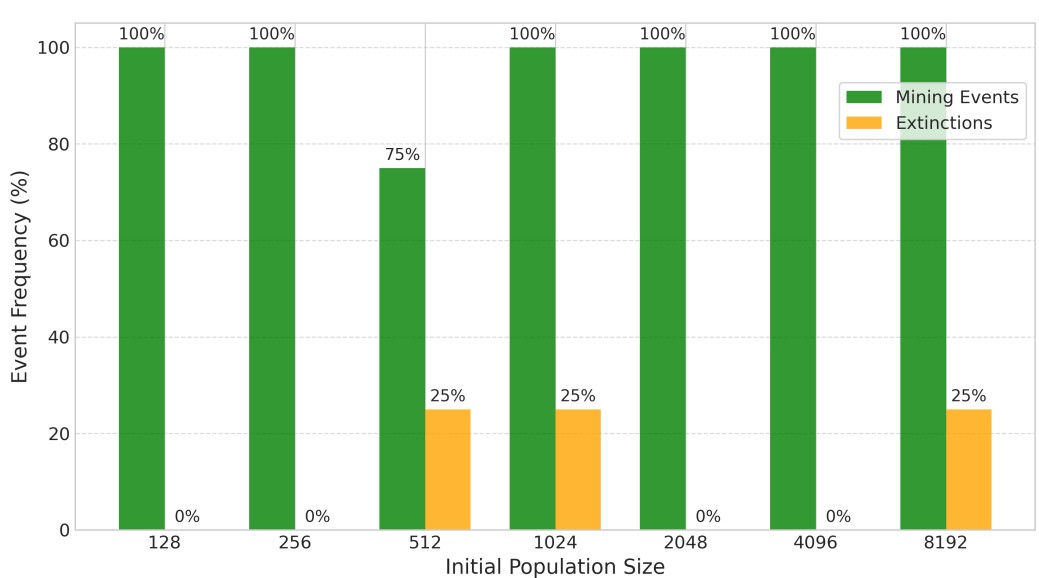

Figure 14: Frequency of mining and extinction events in a $256 \times 256$ **Beach** world for different initial sizes of **(RC)** agent populations. Frequencies are computed across four random seeds for each initial population size; i.e. 25% Extinction Event Frequency means that in 1/4 seeds, the population went to 0 before 2M simulation steps.

Figure 15 provides some insight into why so little effect is visible. Regardless of the size of the initial population, each run rapidly converges to a similar population size in the first 50,000 steps of simulation. This suggests that the environment implicitly specifies a carrying capacity based on the fixed resources available in the system: populations beginning below the carrying capacity grow to reach it, and populations that are too large are constrained to the capacity. As a result, the effective population size is similar after an early correction of the initial population size, leaving mining and extinction frequencies unaffected. Importantly, however, larger initial population sizes contribute somewhat to the total resources in the system because agents carry biomass that is deposited upon death; as a result, larger initial population sizes are able to reach higher maximum populations on average over the course of simulation (Table 9), but the effect is not significant enough to impact mining or extinction frequency for this setting. It is worth noting that although the population size converges early on, it does not stay consistent throughout the run. Later in the simulation, the populations between different runs can vary substantially, as the ecological impacts of agents evolving different abilities at different times changes the later resource distributions.

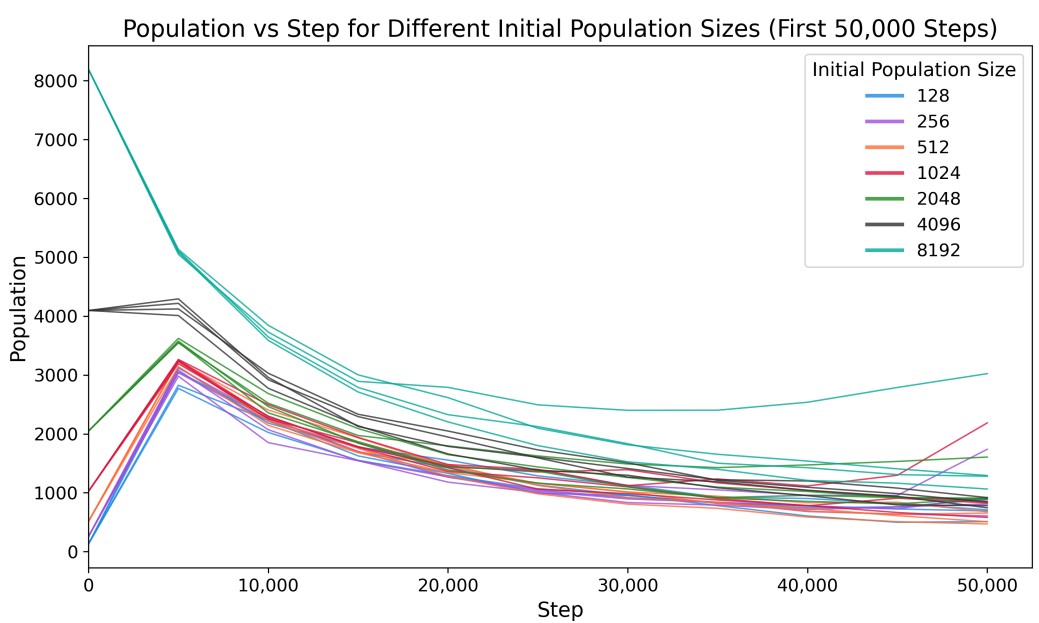

Figure 15: Population size during the first 50,000 time steps in the 256×256 **Beach** with **(RC)** agents, with four seeds per initial population size. The environment supports an implicit carrying capacity based on resources available in the system, and all initial population sizes approach this value early in the run. Larger initial population sizes contribute to the total resources in the system due to biomass deposited on agent death, raising the carrying capacity somewhat.

## F.2 VARYING RESOURCE RULES

We evaluate the effect of two resource hyperparameters, biomass photosynthesis rate and initial biomass site density, on mining and extinction frequency. The setting is equivalent to that in Appendix F.1. The initial population size is fixed at 2048 for this study, and agents once again have compass **(RC)**.

The biomass photosynthesis rate controls the rate at which landscape biomass creates energy via photosynthesis, providing resources for agents over time. The initial biomass site density parameter determines the fraction of grid cells that have biomass at the start of simulation. All experiments in the main body (Section 4) use a biomass photosynthesis rate of 0.01 and an initial biomass site density of 1/4.

While varying initial population size did not affect mining and extinction consistency (Figure 14), changing resource rules has a significant effect as demonstrated in Figures 16 and 17.

Table 9: The maximum population size reached after step 0 for different starting population sizes, with averages and standard deviations across four random seeds. Larger initial population sizes reach slightly higher maximum populations with greater variance across seeds than for lower initial population sizes. However, the max population size reached is largely robust to the initial population size: an initial population size (128) of only 1.6% of an initial size of 8192 reaches 78.9% of the max population reached by the latter (3178 vs. 4026).

| Initial Population Size | Max Population Size After Step 0 |
|---|---|
| 128 | $3178 \pm 217$ |
| 256 | $3056 \pm 61$ |
| 512 | $3451 \pm 437$ |
| 1024 | $3613 \pm 381$ |
| 2048 | $3786 \pm 431$ |
| 4096 | $3689 \pm 437$ |
| 8192 | $4026 \pm 642$ |

Intuitively, larger biomass photosynthesis rates or initial biomass site densities directly boost the effective carrying capacity of the environment by increasing the resources available in the system. As a result, larger populations are supported throughout the run, increasing the likelihood of mutations favoring mining behavior while lowering the chance of extinction. Our data support this intuition: on average (omitting seeds that went extinct), runs with initial biomass site density 1/16 reached a max population of 1608 after step 0, while reaching a mean max population of 2466 for density 1/8, 3786 for density 1/4, and 5353 for density 1/2.

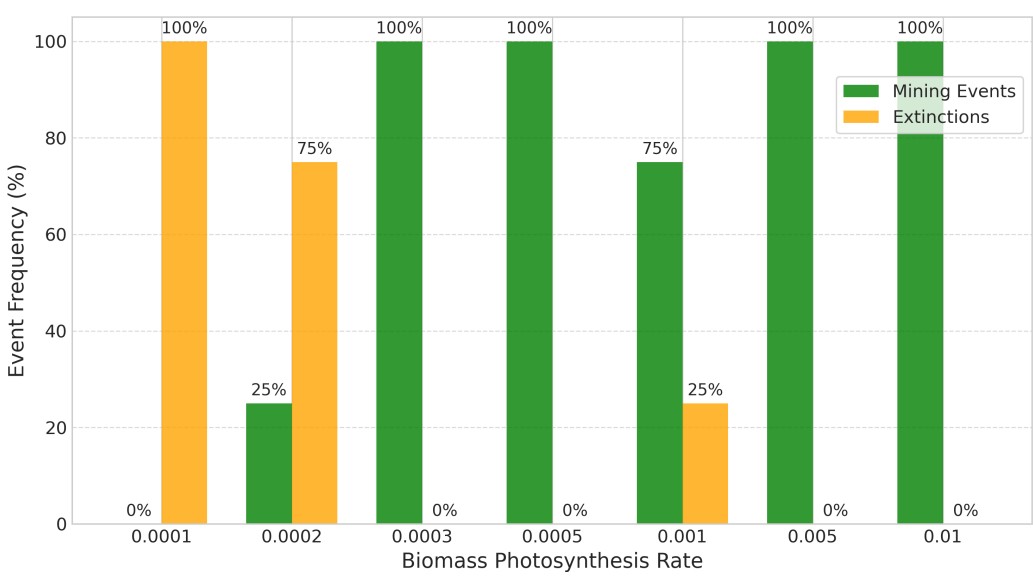

Figure 16: Frequency of mining and extinction events in a 256×256 **Beach** world for different settings of the biomass photosynthesis rate resource parameter. Frequencies are computed across four random seeds for each setting. Mining consistently emerges with biomass photosynthesis rates of 0.0003 and greater.

## F.3 RULING OUT SELF-AVERAGING IN LARGER POPULATIONS

A concern is that the increased stability of mining behavior in large environments (Table 1) might be explained purely by self-averaging rather than by qualitatively different dynamics. By self-

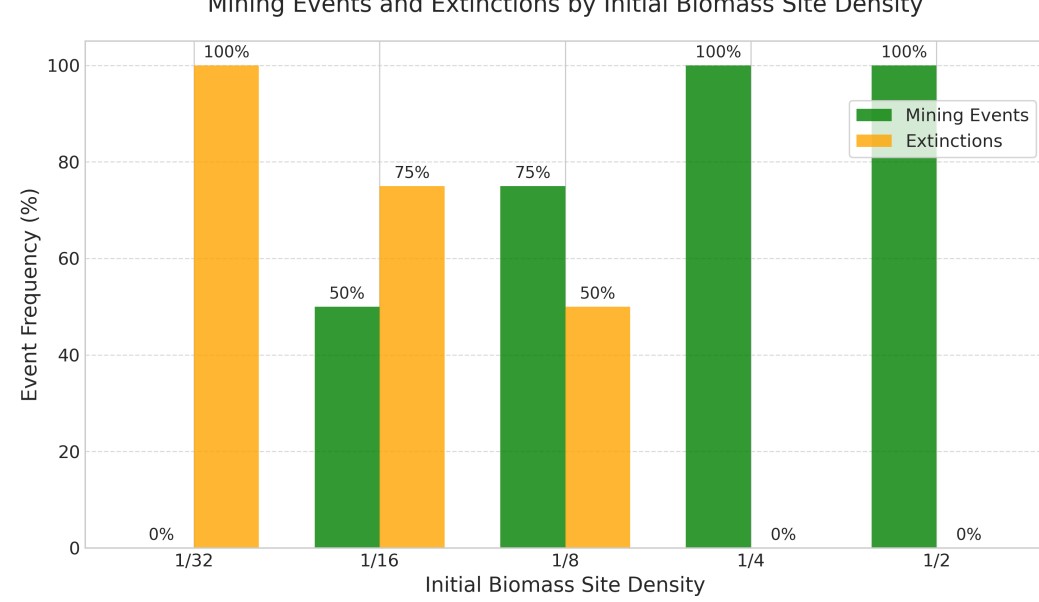

Figure 17: Frequency of mining and extinction events in a $256{\times}256$ **Beach** world for different settings of the initial biomass site density resource parameter. Frequencies are computed across four random seeds for each setting. Greater initial biomass site densities result in more consistent discovery of mining and lower chance of extinction.

averaging, we mean the statistical-physics intuition that, as system size grows, fluctuations in macroscopic quantities shrink roughly as $1/\sqrt{N}$, where $N$ is the system size, such that a single large system behaves like the average of smaller systems of the same total size. Under this view, a $512{\times}512$ **Beach** world might simply be acting like many loosely coupled $128{\times}128$ worlds run in parallel. If that were the case, then our scaling results could in principle be reproduced by running enough small worlds and averaging over random seeds.

To test this hypothesis, we run an experiment holding the total "sample size" roughly fixed while changing how it is partitioned across environments. In the **Beach** terrain with compass-equipped agents (**RC**), we consider three conditions:

- 4 runs of a $512{\times}512$ world with 8192 initial agents each,
- 16 runs of a $256{\times}256$ world with 2048 initial agents each,
- 64 runs of a $128{\times}128$ world with 512 initial agents each.

In each case, the total number of grid cells across runs, the total starting resources and the total initial number of initial agents are identical:

$$4 \times 512^2 \;=\; 16 \times 256^2 \;=\; 64 \times 128^2, \qquad 4 \times 8192 \;=\; 16 \times 2048 \;=\; 64 \times 512.$$

Each seed is run for 2M steps with the same hyperparameters as in Section 4.1. As a control, we repeat this protocol for agents without a compass (**R**). Table 10 summarizes the frequency of long-distance mining and extinction in each configuration.

If large environments were acting as self-averages of many smaller ones, we would expect mining and extinction rates to be roughly invariant to how we distribute grid cells and agents across runs: 4 large worlds or 64 small ones would exhibit similar probabilities of discovering mining. Instead, we observe a strong dependence on world size. For (**RC**) agents, mining emerges in all runs with no extinctions at $512{\times}512$, remains common but not guaranteed at $256{\times}256$, and becomes unreliable and fragile at $128{\times}128$, where many populations go extinct before reaching 2M steps. In contrast, (**R**) agents never develop mining and go extinct at all scales, confirming that the compass remains a key enabler of long-range behavior.

Table 10: Mining and extinction events when trading off world size against the number of independent runs. We compare **Beach** worlds at grid sizes $512 \times 512$, $256 \times 256$, $128 \times 128$ with initial populations scaled proportionally (8192, 2048, 512 agents, respectively). For each size, we allocate the same total number of grid cells and initial agents across all runs by running 4 seeds at $512 \times 512$, 16 seeds at $256 \times 256$, and 64 seeds at $128 \times 128$. We report the fraction of runs in which mining emerged (⛏) and the fraction in which the population went extinct before 2M steps (💀), for agents with a compass **(RC)** and without a compass **(R)**.

| | $512 \times 512$ 8192 | | $256 \times 256$ 2048 | | $128 \times 128$ 512 | |
| --- | --- | --- | --- | --- | --- | --- |
| | ⛏ | 💀 | ⛏ | 💀 | ⛏ | 💀 |
| **(RC)** | 4/4 | 0/4 | 15/16 | 1/16 | 31/64 | 41/64 |
| **(R)** | 0/4 | 4/4 | 0/16 | 16/16 | 0/64 | 64/64 |

These results are inconsistent with a self-averaging explanation. On the contrary, larger environments appear to enable and stabilize mining behavior in ways that cannot be explained solely by averaging over more small simulations.

Even when matching the total number of cells, agents, and simulation steps across conditions, concentrating resources into a few large worlds produces a regime in which mining is consistent and extinctions are rare, while dispersing the same resources across many small worlds gives populations that frequently fail to discover mining and are more likely to collapse. This suggests that increasing environmental scale is not simply smoothing out stochastic noise, but is altering the underlying ecological dynamics, for example by supporting larger, more spatially distributed populations and more persistent diversity in evolving policies.

### F.4  MUTATION RATE SWEEP

We test the sensitivity of the emergence of mining to the mutation rate hyperparameter, which controls the scale of Gaussian noise added to a child agent's policy upon birth. Intuitively, the mutation rate directly impacts how different a child agent is from its parent. We once again run experiments in a $256 \times 256$ **Beach** world with an initial population of 2048 **(RC)** agents for a maximum of 2M steps.

In Figure 18, we show that between mutation rates of $6 \times 10^{-3}$ and $7.875 \times 10^{-2}$, mining emerges in at least 3 of the 4 seeds run, indicating that our results in Section 4.1 with mutation rate $3 \times 10^{-2}$ are robust within this interval. As expected, mutation rates that are too small (e.g., $3 \times 10^{-3}$) prevent agents from developing new behavior and impair survival, leading to extinction. Conversely, large mutation rates (e.g., $3 \times 10^{-1}$) undermine population stability, consistently leading to extinction with no mining adaptation.

### F.5  NETWORK SIZE

We study the effect of the size and architecture of the agent policy network on mining and extinction frequency, as well as the impact of environmental scale on this effect. We investigate $256 \times 256$ and $512 \times 512$ **Beach** worlds with initial population sizes of 2048 and 8192 **(RC)** agents, respectively (scaling by four with the number of grid cells). We ran baselines with **(R)** agents for each experiment, again finding rare to no mining events and 100% extinction rates.

The main experiments in Section 4 used agent policies with 2 layers and 64 channels, yielding a network of 22465 parameters per agent (Table 11). In Figure 19, we show that in the $256 \times 256$ **Beach**, halving channels to 32 (resulting in 9185 total parameters) or doubling to 128 (giving 61313 parameters) with number of layers fixed has minimal to no effect on mining and extinction frequency. Meanwhile, changing network depth has a significant effect: all populations of agents with just 1 layer and 32 channels went extinct, and 50% of populations of agents with 3 layers and 128 channels went extinct.

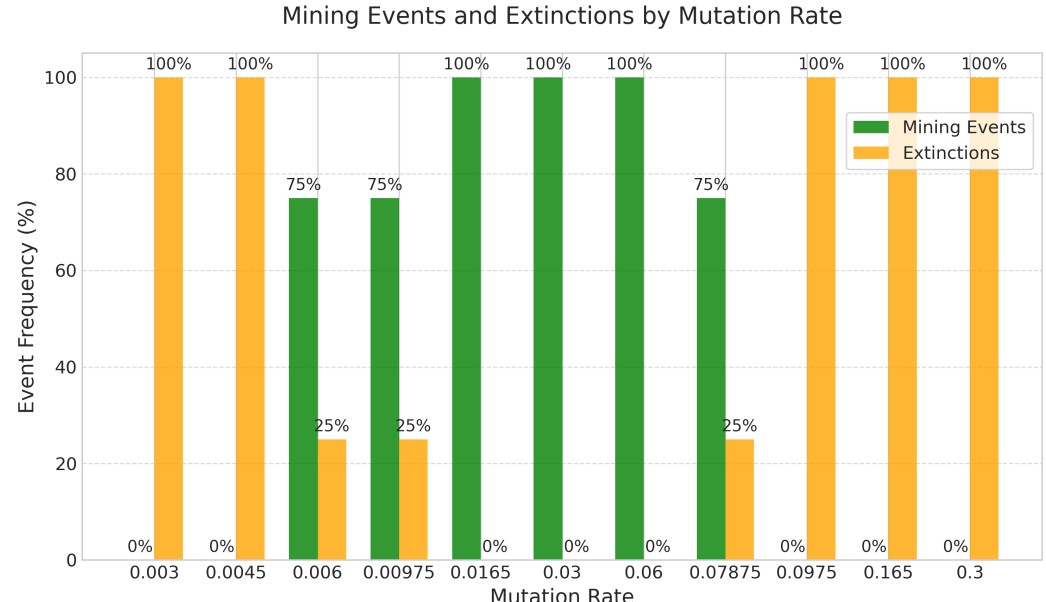

Figure 18: Frequency of mining and extinction events in a 256×256 **Beach** world for different mutation rates. Frequencies are computed across four random seeds for each setting.

Crucially, scaling the environment to the 512×512 grid size produced significantly different dynamics for networks of varying depth (Figure 20). In particular, in the larger world, agent populations with 3 layer, 128 channel networks consistently mined and survived through 2M steps. Surprisingly, agent populations with the small 1 layer, 32 channel networks mined consistently as well. We hypothesize that it is more challenging for both smaller and larger networks to discover new behaviors for different reasons. When the network is smaller, the reduced capacity makes it hard to find good solutions. When the network is larger, the increased size of the search space means it takes longer to find promising new behavior. Regardless of the reasons, the fact that these networks fail so often in small environments but succeed in larger ones further illustrates the stability that comes from increasing scale. The larger environment can support larger populations, which allows for more exploration in policy space. Simultaneously, it takes longer for locally successful mutants to propagate across a large environment, which promotes diversity and again ensures greater exploration. For more discussion of these effects see (Mitchell et al., 2006).

Table 11: Total parameters in **(RC)** and **(R)** agent policies with different network architectures.

| Architecture (layers, channels) | Parameter Count **(RC)** | **(R)** |
|---|---|---|
| 1, 32 | 8129 | 7969 |
| 2, 32 | 9185 | 9025 |
| 2, 64 | 22465 | 22145 |
| 2, 128 | 61313 | 60673 |
| 3, 128 | 77825 | 77185 |

## F.6 Varying Vision Range

We assess the impact of different vision ranges and viewing configurations on predation and foraging behaviors in the 512×512 **Ocean** world with **(RCV+A)** agents. We evaluate visual sensor ranges of 3×3, 7×7, and 9×9. For each range, we evaluate two viewing configurations shown in Figure 21: a) Around: agents can see grid cells all around them b) Front: agents cannot see cells behind them.

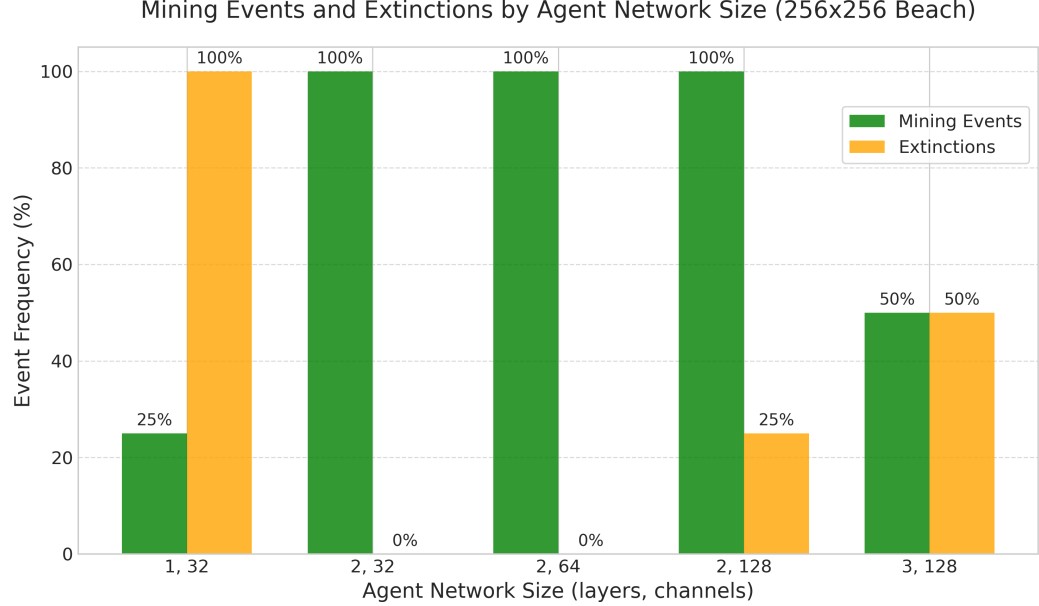

Figure 19: Frequency of mining and extinction events in a $256{\times}256$ **Beach** world for different agent policy network sizes. Frequencies are computed across four random seeds for each setting.

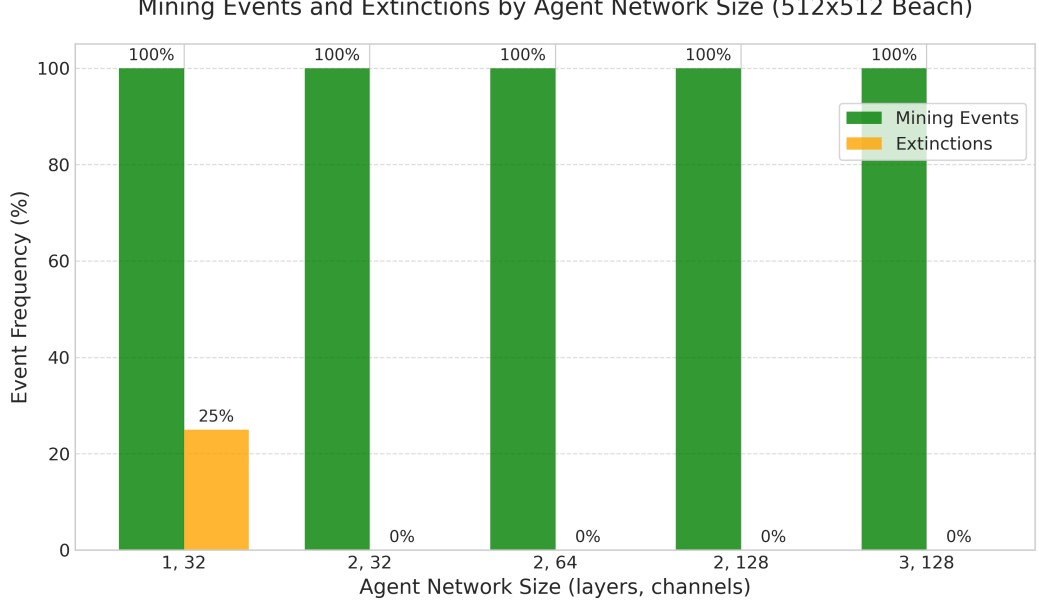

Figure 20: Frequency of mining and extinction events in a $512{\times}512$ **Beach** world for different agent policy network sizes. Frequencies are computed across four random seeds for each setting.

The total number of cells in the agent's vision range is equal for the two configurations. The main experiments with vision in Section 4.2 were run with the $7{\times}7$ Around setting.

These results indicate that a smaller $3 \times 3$ vision range results in behavior that is largely similar to no vision at all (Blind), while the results at $9 \times 9$ appear similar to $7 \times 7$. This indicates that there may be diminishing returns of larger vision ranges in the current environmental configuration. This is intuitive given the rest of the environmental configuration; since the agents have no memory, it is

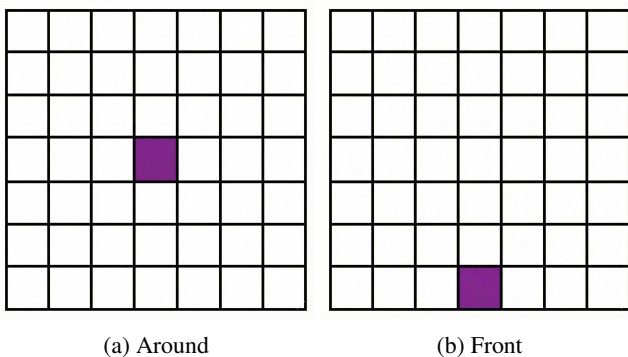

(a) Around          (b) Front

Figure 21: Comparison of the Around and Front viewing configurations for the 7×7 vision range. The purple cell represents an agent, facing the top of the page.

Table 12: Foraging and predation data in the 512×512 **Ocean** world for vision agents **(RCV+A)** with varying vision ranges and viewing configurations. Refer to Tables 5 and 6 for definitions of the metrics represented by each icon. For each seed, the value for each metric was averaged across all steps starting at step 500k to allow them to stabilize. Reported values are the mean across the four seeds and the standard deviation of the four seed averages.

| Vision Range | Viewing Configuration | 👥 | ⚙️ | 🏃 | 😊 | 🗡️ | 🎯 |
|---|---|---|---|---|---|---|---|
| | Blind | $7212 \pm 2496$ | $.181 \pm .060$ | $.435 \pm .090$ | $.392 \pm .033$ | $.032 \pm .006$ | $.307 \pm .090$ |
| 3×3 | Around | $10308 \pm 4016$ | $.258 \pm .105$ | $.352 \pm .014$ | $.427 \pm .022$ | $.040 \pm .030$ | $.388 \pm .195$ |
| | Front | $13863 \pm 520$ | $.351 \pm .023$ | $.336 \pm .008$ | $.437 \pm .009$ | $.026 \pm .006$ | $.470 \pm .145$ |
| 7×7 | Around | $16739 \pm 122$ | $.373 \pm .003$ | $.259 \pm .000$ | $.532 \pm .001$ | $.015 \pm .000$ | $.751 \pm .007$ |
| | Front | $15255 \pm 623$ | $.343 \pm .015$ | $.253 \pm .001$ | $.540 \pm .004$ | $.015 \pm .000$ | $.690 \pm .041$ |
| 9×9 | Around | $16744 \pm 117$ | $.372 \pm .002$ | $.257 \pm .001$ | $.534 \pm .001$ | $.015 \pm .000$ | $.728 \pm .009$ |
| | Front | $15211 \pm 1416$ | $.342 \pm .033$ | $.253 \pm .001$ | $.541 \pm .007$ | $.015 \pm .000$ | $.664 \pm .050$ |

unlikely that visual features at the periphery of a $9 \times 9$ window could meaningfully alter the optimal action relative to a $7 \times 7$ window consistently enough to provide a strong learning signal.

The data also suggest that at the 7×7 and 9×9 vision ranges, the Around configuration is slightly more favorable in terms of population stability and effectiveness of attack and eat actions, resulting in higher average population 👥, biomass utilization ⚙️, and homicides per attack 🎯 than the Front configuration. We hypothesize that this is due to diminishing returns of being able to see more distant pixels; in the Around configuration, more pixels in the agent's vision range are nearby, likely containing more relevant information for near-term actions than distant cells visible with the Front configuration. At the $3 \times 3$ vision range, however, the trend is reversed, with the Front configuration being more favorable than Around; this indicates that in this environment, the marginal benefit of seeing one more pixel ahead through the $3 \times 3$ Front configuration outweighs that of seeing one pixel behind in the $3 \times 3$ Around configuration, which gives most similar behavior to the Blind setting.

### F.7 VARYING ATTACK STRENGTH

We study the effect of reducing agent attack strength on emergent behaviors and population dynamics in the 256×256 **Ocean** world. We evaluate the following attack strengths: 1) Full Attack: deals 10 HP of damage, instantly killing any agent in the attack radius 2) Half Attack: deals 5 HP of damage 3) Quarter Attack: deals 2.5 HP of damage. The main **Ocean** experiments in Section 4.2 were run with the Full Attack setting.

We find that reducing the attack strength from Full Attack to Half Attack significantly reduces the effectiveness and frequency of attacks: attacks per agent and homicides per attack drop considerably (Figure 22). As a result, populations with lower attack strength grow larger, with agents moving and eating more frequently, resulting in greater biomass utilization. Further reducing strength to

the Quarter Attack setting yields diminishing returns in terms of further reduction in violence and facilitation of additional population growth. Intuitively, when attacks do less damage, a homicide requires more attacks per agent, reducing the lethality of each attack. However when the lethality is reduced, attacking behavior becomes less useful and therefore less likely to be selected for in the agent population. Agents with lower attack strengths therefore do not become as skillful at attacking and spend more time foraging for food.

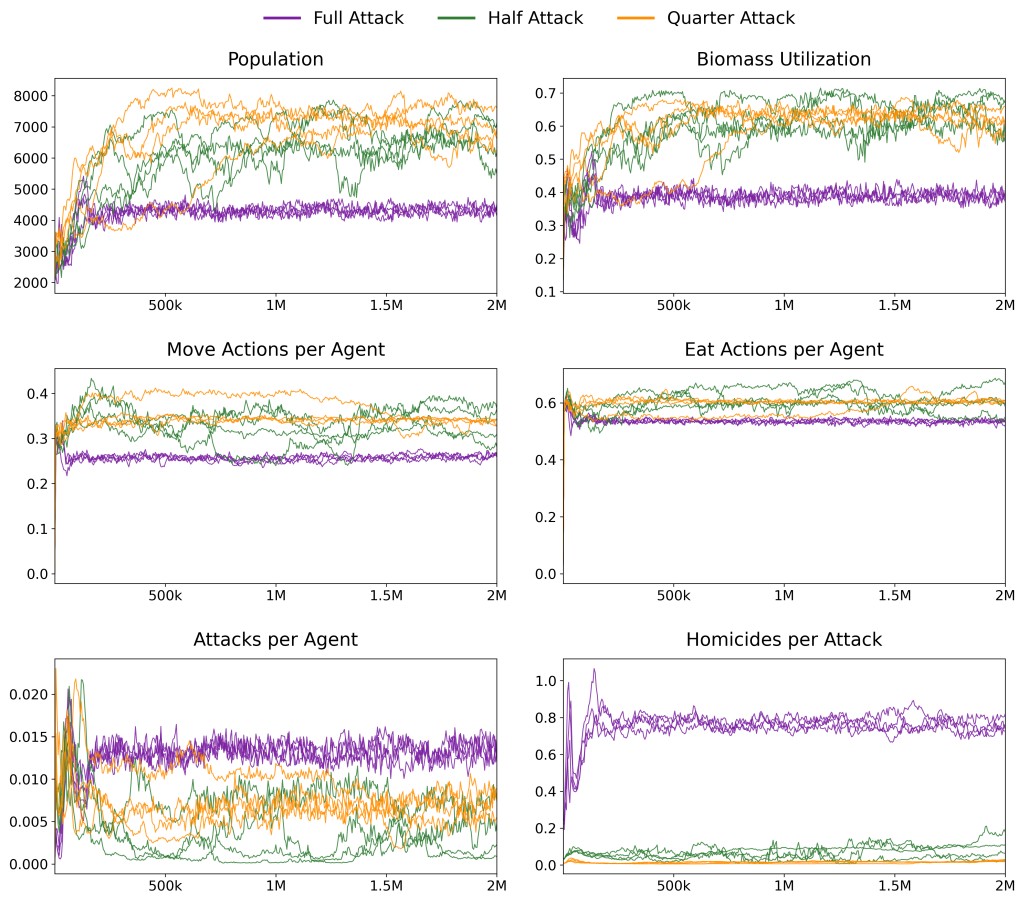

Figure 22: Population and action frequency data over 2M steps in a 256×256 **Ocean** world for **(RCV+A)** agents across Full Attack, Half Attack, and Quarter Attack strength settings. Four seeds are plotted per attack strength setting.

## F.8 VARYING AGENT HP

We investigate the effect of reducing the starting HP of child agents and lowering the healing rate to apply pressure for agents to 'grow' over their lifetime in order to survive. The healing rate parameter determines how much HP an agent can regenerate per healing step, which converts available energy to HP. Agents can influence healing only by maintaining enough energy.

We study two settings: 1) Full HP: child HP starts at 5 and healing rate is 1 HP per 0.1 energy 2) Low HP: child HP starts at 1, with healing rate 0.5. Max agent HP is 10 in all settings. The main experiments (Section 4) were run with the Full HP setting.

Across the two settings, there is no significant difference in homicides per attack, eat action frequency, or biomass utilization (Figure 23). In the Full HP setting, populations grow slightly larger, while in the Low HP setting, agents attack less frequently and move slightly more. We hypothesize that agents born with lower HP must focus on foraging early on in their lifetime in order to grow to full size, which would explain this difference.

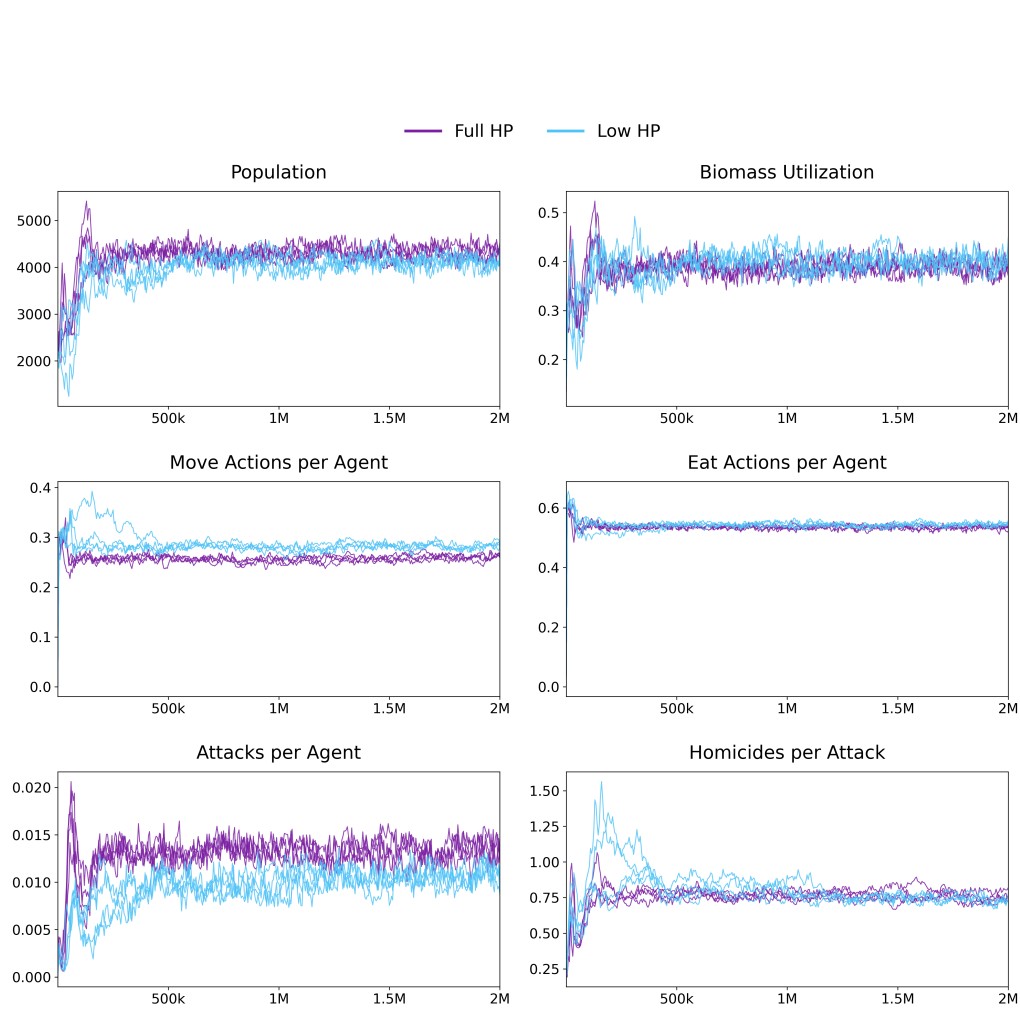

Figure 23: Population and action frequency data over 2M steps in the 256×256 **Ocean** world for **(RCV+A)** agents across Full HP and Low HP settings. Four seeds are plotted per HP setting.

