# OpenReview forum: "The Emergence of Complex Behavior in Large-Scale Ecological Environments"
_ICLR.cc/2026/Conference — Submitted to ICLR 2026_

### Official Review · Reviewer_J3YU · 2025-10-23

**Soundness:** 3
**Presentation:** 4
**Contribution:** 2
**Rating:** 8
**Confidence:** 4

**Summary:**

The authors introduce a large-scale platform for open-ended intelligent agent simulations. In these simulations, large numbers of agents, with their actions decided based on randomly initialized neural networks, can collect resources with which to survive, navigate their environment, attack and kill other agents, reproduce, and, should they run out of health, die. Evolution of the agents is facilitated by the reproduction action, where, assuming an agent has enough resources, it may produce a copy of itself with a small perturbation of the neural network weights, thus creating a simple random search evolution. As a state description, the agents can have access to internal sensors, describing their health, age, and resources, as well as external information, including a compass and local vision. The authors then experiment with allowing the agents access to some or all of this information and different actions, in particular, whether or not the agents can kill, to determine how the populations evolve over time and whether certain information leads to more common policies. They find, for example:

- Agents without a compass appear to die out quickly in environments where land is not surrounded by water. This is due to the land-based agents not finding water, thus dying, and the water-based agents not realizing how to get onto land and collect food before heading back to the water. Given a compass, the agents can learn to "mine" for resources.
- Agents with vision are less likely to kill, but more efficient. Killing in this world involves the agents in front of the attacking agent dying. Therefore, agents with vision are typically better at it.

Overall, the paper is an interesting exploration of emergent policy, although the largest contribution is likely to be the development of the JAX-based environment in which one can run these simulations.

**Strengths:**

The strengths of the paper come in its comprehensive and clear discussion of results and the introduction of a tool that will likely benefit other research efforts. The authors clearly state the arguments behind the behaviors that emerge in their simulations in a way that is easy to understand and of general interest. Further, the development of a JAX-based package to perform further experiments is likely to be of great use to the research community.

**Weaknesses:**

While the paper is well-written and of interest, several weaknesses need to be addressed.

- The experiments in the paper are very limited and likely within the scope of other, similar studies. Therefore, the main result of the paper appears to be the development of the JAX package. To that end, though, there is very limited information provided about the simulation package and its performance. Scaling tests, discussions on GPU deployment, software architecture, and future directions are all absent in a paper where this appears to be a very significant result and contribution to the community.
- The handling of statistics is not very strong. The authors mention that in larger simulations, some of the emergent strategies are more stable. This is likely to be self-averaging of the system with a larger sample size. However, that also means that the smaller simulations were not repeated enough (four times according to the methods) to be fairly compared to the larger ones. A simple solution would be to show the scaling as a function of simulation size to identify saturation and then only present results in the larger runs.
- The argument surrounding non-determinism due to GPU operations could be better explained. Is it a type-casting issue related to the 16-bit weights?
- The absence of captions in Figures 5 and 6, initially assumed to be an oversight, appears now more to be a means of not going over the page limit for the conference submission.

**Questions:**

The following questions came up when reading over the paper.

- Is the vision given to the agents 360 degrees? If so, would it make more sense to only provide limited vision in a cone?
- How are fights decided? From an initial reading, it appears that whoever attacks first wins. It could be interesting to have probabilistic or health-based criteria. How easily can this be implemented and tested in the new framework?
- Further to the previous question, would it make sense to include growth in the agents, perhaps by increasing HP?
- The authors mentioned the color of the agent is changeable. Can the authors elaborate on this point? It doesn't appear to be mentioned further in the work.
- The biggest question that arose was the possibility of guiding the evolution. Simple, random evolution is very limited. How could the authors see including certain drives of the agents, e.g., a desire to live and reproduce, perhaps using limited reinforcement learning algortihms to have a more guided search. Are these different kinds of update strategies simple in the new, JAX-based framework?

**Details Of Ethics Concerns:**

The authors appear to have removed captions to meet the page limit. While minor, it could be seen as a breach of ethics.

---

> ### Author Response · Authors · 2025-11-14
> **Missing captions and page limit**
>
> Thank you so much for the thoughtful review!  A more comprehensive response is coming in a few days, but we wanted to take a moment to first address the issue of the missing captions and potential ethics issue around page count.  The missing captions for Figures 5 and 6 were a genuine mistake and not an attempt to get around page length restrictions.  Had we realized our mistake, we would have ensured the captions were included while still respecting the page limits, which could have been done by shrinking various figures or using brief captions to make space for them in the paper.  We understand how this conclusion could be drawn, and are grateful for the thorough review and attention to detail.  If concerns remain, we would be happy to provide any information necessary to address the issue.

---

> ### Author Response · Authors · 2025-11-22
> **Thank you for your feedback!**
>
> Thank you for your detailed and constructive feedback. We appreciate that you found the work to be well-written, with clear discussions of results, and that the JAX-based simulation platform is likely to benefit the research community.  We will be releasing a revised manuscript soon, but below, we respond to each of your main concerns and questions.
>
> __Limitations of Experiments (First Bullet Under Weaknesses)__
>
> We appreciate this feedback, and have added substantially to the ablations and analyses in the revised draft.  Specifically we will include experiments:
> * Varying the initial population for a fixed map size
> * Varying the network mutation rate
> * Varying the network size
> * Varying the agents’ sensory range
> * Varying the amount of resources in the environment
> * Varying the damage done by attacks
> * Decreasing the agents’ initial HP so that they must grow to full health over time
>
> __Discussion of Simulator (First Bullet Under Weaknesses)__
>
> Thank you for this suggestion, which was echoed by other reviewers.  We will include a section in the revised draft with more details on the simulator’s construction and computational requirements.  To summarize, running one seed for our largest experiments utilizes a single Nvidia H100 for less than one day.  While this hardware is expensive and not available to all researchers, this is much less compute intensive than many language model training runs and within reach of many in the community.  Additionally, our smaller experiments are much more manageable and should be reproducible on consumer hardware in a matter of hours.  Our simulator will be released open-source upon publication.
>
> __Statistics (Second Bullet Under Weaknesses)__
>
> We are currently running experiments to try to determine the extent to which larger environments are merely self-averaging as you say, or exhibit qualitatively different behavior.  We will follow up on this point soon.
>
> __GPU Nondeterminism (Third Bullet Under Weaknesses)__
>
> We appreciate the opportunity to clarify the issues around GPU nondeterminism. It is our understanding that the non-determinism we observe arises primarily from parallel summation ordering and fused floating-point operations.  Essentially, GPUs often change the ordering of operations, which should theoretically be commutative, but in practice can produce subtle differences due to floating point rounding.  These differences compound over time into meaningful drift.  Our understanding is that the 16-bit weights play a role in that they coarsen these rounding issues, but are not the primary cause.
>
> __Captions (Fourth Bullet Under Weaknesses)__
>
> We apologize for the missing captions in Figures 5–6. As explained in our earlier message, this was purely due to a mistake, not an intentional omission to get around page limits.

---

> ### Author Response · Authors · 2025-11-22
> **Continued**
>
> __Vision (First Bullet Under Questions)__
>
> You are correct; the current experiments use 360° vision, but your suggestion of limiting this to a forward facing cone is a good one.  In the new ablations we mentioned earlier, we tested a few alternatives to the agents’ visible range.  We ended up using a square region directly in front of the agent rather than an actual cone for computational reasons, but the effect similarly limits the agents’ ability to see behind them.  Details will be in the upcoming revised draft.
>
> __Fight Rules (Second Bullet Under Questions)__
>
> You are correct that currently the first agent to hit another will kill it instantly, but your suggestion to experiment with alternative arrangements is a good suggestion.  In our new ablations, we experiment with different attack strengths and healing rates.  We do not presently have the ability to connect the attack strength to the attacker’s current health, but may try this towards the end of the review period if we have time.  Details on these experiments are coming soon in the revised draft.
>
> __Agent Growth (Third Bullet Point Under Questions)__
>
> This is another good suggestion.  In the upcoming ablations, we attempt this by making an agent’s starting HP much lower and decreasing the healing rate so that they must grow to full strength over time.  Details on these will also be in the coming revised draft.
>
> __Agent Color (Fourth Bullet Point Under Questions)__
>
> Thank you for pointing out this omission; we forgot to include key details.  As mentioned in the paper, the agents’ color mutates with the rest of their policy.  This color determines what they look like when observed by other agents with the vision sensor.  In theory, this allows other agents to distinguish between different types of other agents, although we do not currently have evidence that they make use of this ability.  We will adjust the upcoming revised draft to make this clear.
>
> __Guided Evolution (Fifth Bullet Point Under Questions)__
>
> Thank you for this suggestion.  While we were not able to provide a reinforcement learning baseline in this paper due to time and space constraints, there is no technical limitation that prevents this within our simulator, which is not strictly limited to the policy mutation approach that we consider here.  Our simulator reports a batch of observations, deaths and reproduction events,  and takes in a batch of actions for each agent at each time step, similar to a gym environment.  This means the simulator itself is agnostic to the policy class and learning rules that map those observations to actions.  We should note that reinforcement learning in large-scale multi-agent settings can be challenging, especially in non-episodic settings like ours with dynamic populations.  With that said, there are some papers, such as the Neural MMO line of work [Suarez et al. ‘19, ‘20, ‘23], that successfully use reinforcement learning in these kinds of settings, albeit with smaller populations than ours.  One important difference, however, is that agents in our simulation only live once, so any updates to their policy must either happen within their individual lifetime, or there must be subpopulations of agents that share a policy and are therefore able to learn from their neighbors’ experiences.  This last possibility is less natural, but because our simulator is agnostic to the policies used to control the agents, it could be supported in our system without modifying the underlying simulation rules.
>
> Thank you again for taking the time to review our work.  Your comments and suggestions have substantially improved the quality of the paper.

---

> ### Comment · Reviewer_J3YU · 2025-11-24
> **Response to author comments**
>
> I would like to thank the authors for their detailed responses. I found them insightful and feel as though they do address the concerns I had when reviewing the manuscript. I believe the initial score stands well and I don't see a reason to change it. As for the concern surrounding the captions, I'm not entirely convinced, but this has not changed the score for the work, so there is no need to address it further.

---

### Official Review · Reviewer_P43N · 2025-10-30

**Soundness:** 3
**Presentation:** 2
**Contribution:** 2
**Rating:** 4
**Confidence:** 3

**Summary:**

This paper examined how environmental scale and population size influence the emergence of complex behaviors in open-ended ecological environments. In this setting, agents have no explicit rewards or learning objectives but evolve over time through life, death, and reproduction rules while continuously shaping their surroundings and populations. The study focuses not on optimizing a single policy but on observing how diverse behaviors, such as long-range resource gathering, vision-based foraging, and predation, emerge from natural competition and environmental pressures. Experiments in large-scale worlds with over 60,000 agents show that some behaviors appear only at larger scales and that increasing scale generally improves the stability and consistency of ecological dynamics.

**Strengths:**

This study clearly shows the originality of the objective-free ecological formulation that couples population dynamics (life/death/reproduction) with partially observable sensing, positioned as a tool to probe emergent behavior at scale. Methodologically, a scalable JAX simulator that supports large maps/populations, explicit resource flows, and controlled sensor ablations, enabling systematic tests of scale and sensing. In the experiments, consistent evidence that compass enables reliable long-range resource trips and vision improves foraging/predation efficiency, with larger worlds reducing variance and increasing stability.

**Weaknesses:**

First, the work relies on mutation-only neuroevolution with memory-less MLPs, but it does not make clear what form of representation or adaptation is actually learned. Second, the paper does not position itself against diversity- and curriculum-driven open-ended learning frameworks, leaving unclear whether the contribution extends or primarily scales existing ideas. Third, the main behavioral effects are intuitive, and because map area and initial population are scaled together, the source of the observed stability (environmental scale vs. population size) remains confounded. For details, see the following questions.

**Questions:**

1. The paper adopts mutation-only neuroevolution with memory-less MLP policies, without any gradient-based optimization or explicit learning objective. Could the authors clarify what kind of representation or adaptation process actually occurs under this setup? For example, how do policy parameters or sensory mappings evolve over time, and can this be interpreted as a form of learning in any representational sense?

2. How does this study relate to unreferenced diversity- and curriculum-driven open-ended learning prior work such as Novelty Search [a], MAP-Elites [b], and POET [c]? A clearer positioning or comparative discussion could help readers understand whether this work extends or merely scales up existing ideas.


3. The results mainly show that a compass leads to navigation and vision improves foraging and predation efficiency, which are intuitively expected outcomes. Beyond the stability that appears at larger scales, what new behavioral or algorithmic insight does this work reveal that was not already known from prior open-ended evolution frameworks?

4. In the scaling experiments, both map area and initial population size increase together.
Could the authors disentangle these two effects (environmental scale vs. population size)?
It may remain unclear which factor actually drives the emergence and stability of complex behaviors.

5. Figures 5 and 6 currently have the “Caption.” only. Please add proper captions.

References
[a] Lehman and Stanley. Abandoning objectives: Evolution through the search for novelty alone.
Evolutionary Computation 19(2), 2011.
[b] Cully et al. Robots that can adapt like animals. Nature 521(7553), 2015.
[c] Wang, Rui, Joel Lehman, Jeff Clune, and Kenneth O. Stanley. POET: Paired Open-Ended Trailblazer.
GECCO 2019.

---

> ### Author Response · Authors · 2025-11-22
> **Thank you for your feedback!**
>
> Thank you for the thoughtful and constructive feedback. We are encouraged by the recognition of our study’s originality in coupling population dynamics with partially observable sensing, as well as the strength of our scalable JAX simulator and empirical evidence for emergent behaviors at scale. A revised draft will be posted soon, but below, we address specific questions and concerns.
>
> __Mutation and Memory (Weaknesses “First, …” and Question 1.)__
>
> Thank you for the opportunity to clarify this.  You are correct that we use only mutation without any gradients.  This means that agent policies only change between generations and do not change over their lifetime.  Consequently, there are two ways that the environment can impact the long-term adaptation of agents within the system:  First, if an agent’s policy is not as good at finding resources as other members of its immediate community, it may starve before it has a chance to reproduce, and therefore its suboptimal policy will be removed from the broader population.  Secondly, if an agent’s policy is good at collecting resources, it may be able to reproduce many more times than other members of its immediate community, which means that this more successful policy will have more copies in future generations, but also that more exploration will be done around that successful policy due to the mutations each of its children will receive.  In these ways, the population will gradually remove policies that are unable to compete with neighbors for resources and promote those that are better able to gather resources and reproduce.  In an extreme example of this, the agents in the Beach environment in Section 4.1 that are first able to make the round trip from water to land and back again suddenly have access to a rich trove of resources that their neighbors cannot access.  This allows them to rapidly grow in number and spawn offspring that can explore for longer distances onto the land.
>
> __Relationship to Diversity and Curriculum based Learning (Weaknesses “Second, …” and Question 2.)__
>
> Thank you for the opportunity to discuss our relationship to these important works. Your concerns about the positioning of this work was shared by reviewer hMQJ.  MAP-Elites and POET are both approaches designed to use diversity and curricula to find good solutions to a particular objective.  In contrast, we are not trying to find solutions to the “problem” of our environment but rather to better understand how environmental and sensing parameters in the environment affect the emergence of complex behaviors.  Novelty Search is more similar in that it also does not have direct objectives, but the high-level goal is still different.  Novelty Search attempts to find as many diverse candidate solutions as possible, while our objective is to study the relationship between environmental parameters, sensing and the emergence of complex behavior.  Based on this feedback and that of other reviewers, we are substantially revising the related work and the discussion of our positioning in other sections, which will be available in the revised draft.
>
> __Behavioral and Algorithmic Insight (Weaknesses “Third, …” and Question 3.)__
>
> While compass-based navigation and vision-based foraging are intuitive results based on our environments, our findings go beyond verifying expected outcomes.  Our experiments show that mutation and competition-based population dynamics alone are enough to produce complex behavior in large populations of agents controlled by deep neural networks.  Beyond this, our work also serves as a counterpoint to the folk wisdom that evolutionary learning is too slow and unwieldy when working with deep neural networks.  The fact that the complex behaviors shown here can emerge from very simple evolutionary rules and population dynamics in a few hours on a modern GPU demonstrates that further exploration of this space is easier than commonly expected.  While other studies such as [Lu et al. 24, Hamon et al. 23] have demonstrated some limited emergent behavior from similar, but much smaller environments and populations, we believe that our larger scale and breadth of experiments (especially after the addition of the new ablations mentioned below) more conclusively demonstrate the potential of ecologically grounded open-ended learning.
>
> [Lu et al. 24] “JaxLife: An Open-Ended Agentic Simulator”
>
> [Hamon et al. 23] “Eco-evolutionary dynamics of non-episodic neuroevolution in large multi-agent environments”

---

> ### Author Response · Authors · 2025-11-22
> **Continued**
>
> __Initial Population (Weakness “Third, …” and Question 4.)__
>
> Thanks for the chance to clarify this.  As we will show in new ablations in the revised draft, the size of the initial population does not strongly impact the behavioral complexity that emerges later.  This is because our environment supports a dynamic population, which means that the agents will quickly grow to an approximate carrying capacity based on the resource availability.  We have also conducted several other ablations (see below), including ones that reduce the total resource distribution, which has a much stronger impact on the overall population size and the emergence of various behaviors, but will save details on this for the revised draft.
>
> __Mission Captions (Question 5.)__
>
> We apologize for the missing captions; they will be present in the upcoming revised draft.
>
> __Additional Comments__
> In the revised draft we are including a number of additional ablations based on the feedback from multiple reviewers.  These aim to disentangle the effects of many of the design decision of our environment and policy networks, including:
> * Varying the initial population for a fixed map size
> * Varying the network mutation rate
> * Varying the network size
> * Varying the agents’ sensory range
> * Varying the amount of resources in the environment
> * Varying the damage done by attacks
> * Decreasing the agents’ initial HP so that they must grow to full health over time
>
> Details of these ablations will be available in the upcoming revised draft.
>
> Thank you again for your detailed feedback and suggestions, your input has substantially improved the quality of the paper.

---

### Official Review · Reviewer_NFvD · 2025-10-31

**Soundness:** 3
**Presentation:** 3
**Contribution:** 2
**Rating:** 6
**Confidence:** 3

**Summary:**

This paper presents a large-scale ecological simulation in which tens of thousands of neural agents live, eat, reproduce, and die without any explicit reward or predefined objective. Each agent’s behavior is governed by a small feedforward neural network whose weights are inherited and randomly mutated during reproduction, forming a purely evolutionary learning process. By systematically scaling the ecological environment—world size, terrain, population, and sensor richness—the authors show that increasingly complex behaviors (such as long-distance resource collection, navigation, and vision-based predation) emerge reliably only at large scales. The work argues that ecological scale can serve as a new axis of capability emergence, much like model scale in modern deep learning. This paper is novel and inspiring, presenting an large-scale ecological simulation that explores how complex behaviors can emerge from simple evolutionary rules without any explicit reward. That said, the work is still more exploratory than analytical—its claims about emergence rely mainly on qualitative observation, and the mechanisms behind the behaviors are not deeply quantified or explained.

**Strengths:**

1. Fresh perspective on open-ended learning. The paper reframes “intelligence without reward” as a scalable ecological process. The analogy between ecological and model scaling is elegant and offers a new lens on emergence in machine learning.

2. The authors manage to simulate up to 60 000 agents in a large heterogeneous world with realistic resource flow, physics, and reproduction—all efficiently implemented in JAX.

3. Convincing qualitative behaviors. Emergent patterns—migratory resource transport, coordinated foraging, predation—appear genuinely spontaneous. The controlled scaling studies (terrain, sensory modalities, map size) clearly show that these behaviors depend on environmental complexity.

4. Interdisciplinary impact. The work bridges artificial life, multi-agent systems, and open-ended evolution.

**Weaknesses:**

1. Lack of quantitative behavioral metrics. Claims about “emergence” are supported mainly by qualitative observation. There are no explicit metrics for behavioral diversity, complexity, or ecological stability.

2. Minimal evolutionary mechanism. The use of pure mutation without crossover or explicit selection limits interpretability of the evolutionary dynamics. It’s unclear whether complexity arises from environment pressure alone or random drift.

3. Missing ablations and baselines. The paper would be stronger if it included smaller-scale or simplified control experiments to isolate the effect of each design choice (e.g., mutation variance, resource rules, sensory range).

4. Compute intensity and reproducibility. Running such large worlds likely requires significant hardware. The paper does not discuss runtime or resource requirements, which may hinder reproducibility.

**Questions:**

1. Can you provide a quantitative measure (even heuristic) for behavioral diversity or ecological balance over time?

2. How sensitive are the observed behaviors to mutation rate or neural network size?

3. Are there global population caps or only local resource constraints?



4. Have you considered hybrid models that combine within-lifetime learning (e.g., gradient updates) with across-generation evolution?

---

> ### Author Response · Authors · 2025-11-22
> **Thank you for your feedback!**
>
> Thank you for the thoughtful and encouraging assessment of our work, particularly the recognition of its novelty, interdisciplinary relevance, and the emergent complexity demonstrated in our large-scale ecological simulations. We are producing a revised draft that will be available soon, but address the main concerns below.
>
> __Behavioral Metrics (Weakness 1)__
>
> We appreciate this suggestion and agree that better quantitative measures would complement qualitative observations.  In the revised draft, we will include additional experiments aimed at addressing the questions around behavioral diversity, complexity and ecological stability.  We are aiming to cluster agents into lineages by means of numerical marker traits analogous to genetic barcodes and make quantitative comparisons between these evolved lineages. For example, we intend to compare the action distributions demonstrating unique strategies between evolved clusters, rates of reproduction, and population size. Diversity can be measured by the number and distribution of clusters, each of which occupy a distinct ecological niche, and stability can be measured as the structure of these clusters over time.  These experiments are not yet complete, and so may differ slightly from what is described here, but details will be available in the upcoming revised draft.
>
> __Evolutionary Mechanism (Weakness 2)__
>
> Thank you for this feedback.  Our intention was to isolate the role of environmental scale and sensing in driving behavioral complexity using a very simple evolutionary mechanism (mutation only). We agree that adding crossover or explicit selection could enrich the dynamics, but our goal was not to simulate true genetics, but instead to explore the kinds of behavior that emerge from very simple update rules and show that complex strategies can emerge even in very simple settings. Stated another way, we feel that the lack of sophisticated rules is a feature and not a bug, because we are not attempting to produce the most complex behaviors using as many evolutionary features as possible, but instead show that complex behaviors can emerge even from simple rules and study the impacts of environmental parameters and sensing modalities on these emergent behaviors.  However, we do agree that this point could be made more clearly and are updating the revised draft accordingly.
>
> On the topic of environmental pressure vs. random drift, we believe that our Beach examples make a strong case for environmental pressures impacting the quality and distribution of agent policies in our environments.  Before the agents have evolved long-distance resource mining, the population steadily declines due to resources being lost as agents accidentally wander onto land and die of thirst (approximately the first 100k steps of Figure 4).  The resources from these agents accumulate on land and cannot yet be effectively recovered, as agents have not evolved behavioral policies able to make the round trip from the water, onto land and back again.  As agents begin to evolve the ability to make this round trip, everything changes.  At first, only a small subset of the population is engaged in this behavior (Figure 4, inset b), but soon a large portion of the population is making long journeys onto dry land and reaching the farthest resources (Figure 3, inset c).  While these policies were discovered using only random mutation, the pressure from the environment, in the form diminishing resources in the water, and the opportunity of a rich basket of resources just out of reach, has clearly impacted the success of these policies.  In contrast, this concentration of long-distance resource gathering policies does not develop in the ocean environment where resources are more evenly distributed, offering further evidence that such behaviors develop in response to environmental pressures.  We will attempt to make this more clear by comparing the differences between evolved policies in the ocean and beach environments in the revised draft.
>
> __Ablations and Baselines (Weaknesses 3)__
>
> Thank you for the note about ablations, this was a concern shared by the other reviewers.  We have added a number of new ablations to show sensitivity to various environmental and evolution parameters.  Details will be in the upcoming revision, but in summary, the updated ablations are:
> * Varying the initial population for a fixed map size
> * Varying the network mutation rate
> * Varying the network size
> * Varying the agents’ sensory range
> * Varying the amount of resources in the environment
> * Varying the damage done by attacks
> * Decreasing the agents’ initial HP so that they must grow to full health over time

---

> ### Author Response · Authors · 2025-11-22
> **Continued**
>
> __Computational Efficiency and Reproducibility (Weaknesses 4)__
>
> Thank you for this note, we are adding a new reproducibility appendix detailing hardware, runtime, and scaling behavior.  To summarize, running one seed for our largest experiments utilizes a single Nvidia H100 for less than one day.  While this hardware is expensive and not available to all researchers, this is far less compute intensive than many language model training runs and within reach of many in the community.  Additionally, our smaller experiments are much more manageable and should be reproducible on consumer hardware in a matter of hours.  Additionally, all of our code will be released open-source upon publication.
>
> __Behavioral Diversity and Ecological Balance (Question 1)__
>
> Thank you for this helpful suggestion.  As mentioned in “Behavioral Metrics (Weaknesses 1)” above, we are adding new experiments in an attempt to disentangle this.
>
> __Sensitivity to Mutation Rate and Neural Network Size (Question 2)__
>
> This is also a great suggestion.  As mentioned in “Ablations and Baselines (Weaknesses 3)” above, we are running additional experiments to analyze this.  We will discuss these results more thoroughly in the upcoming revised draft, but to summarize briefly: we find that there is a reasonable range of mutation rates that produce broadly similar emergent behavior, but as might be expected, if the mutation rate is too low, agents are not able to meaningfully adapt relative to their parents, and if it is too high, the resulting behaviors become unstable.  Similarly we find that there are a number of network configurations that produce similar behaviors, but there are networks that are too small to produce some of them.  Interestingly, it appears that if the networks are too large, this can also hinder evolutionary emergence, as this increases the evolutionary search space.  We do find, however, that these larger networks can still discover the same novel behaviors in larger environments, which allow for more exploration.  Details of these experiments and analysis will be provided in the revised draft that will be posted soon.
>
> __Population Caps (Question 3)__
>
> For technical reasons, our environment does require a maximum population to be specified as a configuration parameter for each experiment in order to allocate memory for the agents.  However, in practice we always set this maximum value to be higher than what we expect to encounter in the actual experiments.  In cases where this estimation was wrong, and we did end up hitting the maximum population cap, we simply increased the maximum population cap and reran the full experiment.  To summarize, while a population cap does exist technically for computational reasons, the cap never restricted actual populations in any of our published experiments.
>
> __Within Lifetime Learning (Question 4)__
>
> Thank you for this suggestion.  We are interested in this as a line of future work.  This adds some implementation complexity, but is definitely something we are interested in pursuing.
>
> Thank you again for your detailed comments and suggestions.  This feedback has substantially improved the paper, and we appreciate your input!

---

### Official Review · Reviewer_ssLs · 2025-10-31

**Soundness:** 2
**Presentation:** 3
**Contribution:** 1
**Rating:** 2
**Confidence:** 4

**Summary:**

The paper seeks to understand how complex behaviors can emerge through large-scale ecological environments with agents (on the order of 10,000s) interact with each other and the dynamic and changing environment. The paper describes how the environment and agents work, and discusses what happens during the evolution of the agents with different environment settings.

**Strengths:**

- The paper introduces a jax based environment which is easy to scale on a GPU/multi-GPU setup to enable faster data generation for studying emergence.
- Environment looks like something that can be easily visualized which is a great for developers/researchers in the future.

**Weaknesses:**

- Overall I feel this paper lacks significantly novelty. Perhaps one novelty is that this is a large-scale environment with 60,000 agents with some interesting game rules (although they seem similar to neural MMO e.g. foraging). Another concern is that the majority of this paper is spent 1. explaining the game and rules followed by 2. analyzing what happens if you evolve all these different (memoryless) agents and what behaviors emerge. There is not much discussion around how these results would teach us anything new about how complex behaviors emerge as a result of scale. As a result this work seems otherwise very similar to the hide and seek open AI work or neural MMO itself.
- A large concern is that the agents are all memoryless, which appears to be a significant drawback when comparing to e.g. neural MMO which uses RNN based agents. Memory is a fundamental part of human behavior and not having this capability as a baseline or approach in the environment would limit the possibilities of the proposed environment.
- Figure 5 missing caption

**Questions:**

- It is interesting that RL is not considered and only population evolution operations are performed by adding noise to parent descended weights. What is the reasoning behind this?

---

> ### Author Response · Authors · 2025-11-22
> **Thank you for your feedback!**
>
> Thank you for the detailed feedback and for recognizing the scalability and visualization strengths of our environment. We are compiling a revised draft which will be available soon, but address the specific questions and concerns point by point below.
>
> __Novelty__
>
> We regret not making the novelty of our work more clear.  As you mention, our experiments allow for much larger populations than have been possible in the past.  This allows us to explore the impacts of population scale on the emergence of complex behavior in a way that was not possible previously.  You mention Neural MMO and the hide and seek work.  These environments include more complex dynamics than our system but use reinforcement learning to provide a direct learning signal to agents.  In contrast to these settings, our agents evolve purely from mutation and population dynamics without any direct reward or other learning signal.  Additionally, the populations in these settings are much smaller than ours and do not support reproduction, limiting their suitability for studying ecological dynamics.   While in some cases the agents in these prior works may demonstrate more complex behaviors, they do so with a much stronger learning objective.  Our goal was not to improve behavioral capabilities beyond what was demonstrated in these previous works, but instead to show that complex behavior can emerge even in simple memoryless agents from very weak evolutionary signals, and investigate the extent to which environmental parameters and sensory abilities impact the emergence of these behaviors.
>
> There are other recent papers such as [Lu et al. 24, Hamon et al. 23] which explore behavioral emergence in a reward free setting similar to ours, however their experiments are somewhat limited in terms of both scale and experimental exploration.  From a scale perspective, both of these deal with populations of only a few hundred or few thousand agents, where we reach populations of tens of thousands, an increase of one to two orders of magnitude.  This gives us much more room to explore the importance of scale as a factor in the resulting emergent behavior.  In terms of experimental exploration, these other papers also provide only a limited number of experiments with few controls, where we compare multiple environments and sensor configurations.  A new set of ablations (see below) we are currently running for the discussion period increases the extent of these even further.
> Thank you for this note, and the opportunity to clarify our novelty and contributions, revisiting this has made the paper much more successful.
>
> __Memory__
>
> Our choice for using memoryless agents was deliberate for two reasons: First, we wanted to avoid complexity and reduce the parameters of each agent’s neural network in order to maximize the population size we could handle in experiments.  Second, our goal was not to improve behavioral performance relative to some previous metric, but instead find out the extent to which complex behaviors can still develop in spite of the limited environmental dynamics and small network size.
>
> Although the agents do not have explicit memory, their internal resources can in some cases be thought of as a very small and primitive form of memory.  For example when the agents learn to mine resources on dry land, their decision to go further inland or turn around and head back to the ocean depends on how much resources they are currently carrying.  In this sense, the amount of water they currently have acts as a very primitive memory of how long it has been since the agent visited the ocean and had a drink.
> We regret that these points were not made in the submitted draft, and are updating the revision to make this more clear.  Furthermore, we intend to extend our work to memory-based agents in future work.
>
> __Missing Captions__
>
> Thank you for bringing this to our attention. We apologize for the missing captions and will address this in the upcoming revision.
>
> __Reinforcement Learning__
>
> Our goal is to explore the emergence of complex behaviors in response to mutation and competitive pressure from other agents instead of using explicit rewards and reinforcement learning as has been done in Neural MMO, for example.  We made this decision in order to reduce the problem of behavioral emergence to as simple a framing as possible, allowing large scale experiments and computational tractability.

---

> ### Author Response · Authors · 2025-11-22
> **Continued**
>
> __Additional Comments__
>
> In the revised draft we are including a number of additional ablations based on the feedback from multiple reviewers.  These aim to disentangle the effects of many of the design decision of our environment and policy networks, including:
> * Varying the initial population for a fixed map size
> * Varying the network mutation rate
> * Varying the network size
> * Varying the agents’ sensory range
> * Varying the amount of resources in the environment
> * Varying the damage done by attacks
> * Decreasing the agents’ initial HP so that they must grow to full health over time
> * Details of these ablations will be available in the upcoming revised draft.
>
> Thank you again for your detailed feedback.  These comments have substantially improved the paper, and we appreciate your input.

---

> > ### Comment · Reviewer_ssLs · 2025-11-22
> > **Response**
> >
> > Thanks for answering my questions.
> >
> > I’m still concerned regarding the lack of analysis of the new scale you have introduced (number of individual units).
> >
> > At most there is Table 1 but it appears to show that at the 3 largest levels of population scales the behaviors learned are the same for the RC agent. This suggests that this environment may not be useful enough to study what happens/emerges when there are 60,000 units compared to 2000 as there is not many differences in results.
> >
> > There does seem to be some difference occurring between a starting population of 32K vs 8192 for the R agent, but then again insufficient analysis for explaining away confounding variables like map resource distribution (as a function of map size)
> >
> > I will reserve further judgement until the new analyses you have mentioned are completed. At first glance they seem to be experiments that should have been done earlier (meaning this paper is unpolished), but they also may answer many of my confusions around confounding factors too.

---

### Official Review · Reviewer_hMQJ · 2025-11-10

**Soundness:** 3
**Presentation:** 2
**Contribution:** 2
**Rating:** 2
**Confidence:** 4

**Summary:**

This paper uses a new JAX-based simulation environment the authors developed to allow rapid evaluation of large grid worlds, where they conduct experiments with populations of more than 60,000 individual agents, each with their own evolved neural network policy. They report on their experiments.

Their stated aim is "to examine how behaviors emerge and evolve across large populations due to natural competition and environmental pressures." They observe some behaviours they refer to as emergent, that arise more commonly with scale. They do not however discuss if similar behaviors have or have not been observed in previous simulations in the literature. The computing framework they develop to enable their simulations is not positioned as their central contribution, and the design is not discussed in detail nor compared to that of other simulations, but it's possible that some novelty of their work lies partly in that HPC contribution.

**Strengths:**

The authors developed a new simulation environment to allow rapid evaluation of large grid worlds, where they conduct experiments with populations of more than 60,000 individual agents, each with their own evolved neural network policy. They observe some behaviours they refer to as emergent, that arise more commonly with scale. Specifically these are: the ability of agents to travel long distances inland in order to find resources (and in fact to go back and forth to water, a behaviour they call "mining") and the ability to use vision sensors to effectively locate free resources or prey on other agents.

The topic in ecology and study of emergence in collections of agents is very nice and interesting.

The effort to set up a platform where such large-scale studies can be carried out is a strength. It would be good to see more information about how the computational design in this paper advances state of the art.

**Weaknesses:**

Contextualization with respect to other work is insufficient.

In particular, there is no information in the experimental results discussion about how such observations compare to ones in related work. This would be important if this paper is meant as a contribution on the (computational) ecology side, but it's also crucual in order to assess the strength of their simulation, e.g. it allows them to remedy specific weaknesses in previous work. The Related Work section near the start of the paper, which does list extensive related literature, also doesn't position the authors' contribution with respect to those other works, it simply lists them. Perhaps the largest contribution of the paper is on the high-performance computing side, i.e. the new JAX-based design they propose. That HPC work is however not stated as the paper's focus and there are few details of that design.

There are also presentation issues, and instances where rigor is lacking:

Re Section 3.1:
- it sounds like in an EG each agent has full knowledge of the id’s of other living agents, since this information is  encoded in s_t… is that really what you assume? it’s quite different from more evolutionary settings.. Michael Levin for example emphasizes the role of information transfer in self-organization (via bioelectiricity).
- Line 164 has sloppy wording: do you mean "it maps for each player, a state-action pair (s_t, a_t) to a distribution…"
- Definition of Gamma: this should be spelled out more clearly.. \Gamma(s_t) is a list of agents alive at time t and for each of them the
identify of their parents.
- In which case, why do you nead G_t? isn’t this information included in \Gamma(s_t)?

Re Section 3.2:
Rigor of this section is important to understand the paper, and for any comparison with learning paradigms like RL or genetic algorithms but it's too informally worded.
- line 174: the notation s_t = {s^m_t, s^a_t} suggests ignorance of the mathematical meaning of these symbols and also goes against convention to call a state space by an upper case letter, instead this is some hybrid between trying to say what each state is, and discussing state space. It seems you’re saying that each s_t = (s_t^m, s_t^a)--notice the parentheses not curly brackets--in which case the state space is S_t = S_t^m x S_t^a. Or maybe the intent is something else. It should be rigorously stated.
 - line 185ff, make clear which factors of S_t^m are changing with time and which are unchanging.
- line 199ff, how much  memory does each agent have? what else governs agents? you need a section for agents
- throughout this section, there are many assumptions.. how would changing these affect the outcome? no ablation analysis was done..

**Questions:**

What would the authors say is their central contribution and how does it advance state of the art, i.e. what exactly does it do *better than comparable* other approaches or past work?

---

> ### Author Response · Authors · 2025-11-22
> **Thank you for your feedback!**
>
> Thank you for your thoughtful and detailed comments. We especially appreciate the constructive feedback and have substantially revised our paper to address the concerns raised. Below, we clarify our main contributions, expand on contextualization and rigor, and provide specific revisions addressing the points made in the review.
>
> __Positioning and Context (Weaknesses)__
>
> We agree that the positioning of our main contributions could be more clear.  In an attempt to improve this, we can consider our paper relative to two specific categories of related work: papers that explore emergent behavior using reinforcement learning or some other supervision signal, and work that explores emergent behavior relying on mutation and population dynamics alone.
> In the first category, there have been a number of successful papers that have shown emergence of complex behaviors in response to reinforcement learning objectives, for example learning to construct hiding places in a multi-player hide-and-seek game [Baker et al. ‘19], learning to adapt to novel environments [Open-Ended Learning Team ‘21], and learning cooperation and role specialization in a large multiplayer game [Suarez et al. ‘19, ‘20, ‘23].  In contrast to these settings, our agents evolve purely from mutation and population dynamics without any direct reward or other learning signal.  Additionally, the populations in these settings are much smaller than ours and do not support reproduction, limiting their suitability for studying ecological dynamics.  While in some cases the agents in these prior works may demonstrate more complex behaviors, they do so with a much stronger learning objective.  Our goal was not to improve behavioral capabilities beyond what was demonstrated in these previous works, but instead to show that complex behavior can emerge even in simple memoryless agents from very weak evolutionary signals, and investigate the extent to which environmental parameters and sensory abilities impact the emergence of these behaviors.
>
> In the second category, there are other recent papers such as [Lu et al. 24, Hamon et al. 23] which explore behavioral emergence in a reward free setting similar to ours, however their experiments are somewhat limited in terms of both scale and experimental exploration.  From a scale perspective, both of these deal with populations of only a few hundred or few thousand agents, where we reach populations of tens of thousands, an increase of one to two orders of magnitude.  This gives us much more room to explore the importance of scale as a factor in the resulting emergent behavior.  In terms of experimental exploration, these other papers also provide only a limited number of experiments, with few controls, where we compare multiple environments and sensor configurations.  A new set of ablations (see below) we are currently running for the discussion period increases the extent of these even further.
>
> With this in mind our primary contributions are:
> * To our knowledge, our experiments contain some of the largest simultaneous populations of agents controlled by deep neural networks to date.
> * This scale gives us the unprecedented ability to examine the effects of physical size and population on the emergence of complex behavior due to evolutionary response to ecological pressure.
> * We show that increasing scale produces new emergent behavior more reliably, and demonstrate the effects of different sensor modalities on the emergence of these behaviors.
>
> Beyond this, our work also serves as a counterpoint to the folk wisdom that evolutionary learning is too slow and unwieldy when working with deep neural networks.  The fact that the complex behaviors shown here can emerge from very simple evolutionary rules and population dynamics in a few hours on a modern GPU demonstrates that further exploration of this space is easier than commonly expected.
>
> To reiterate, we agree that this framing was not made clear in our initial draft, and have edited the abstract, introduction, related work and conclusion to more clearly state our contributions and framing relative to existing literature.  This will be available shortly in a new revised draft.  Thank you again for your constructive feedback and for drawing our attention to this lack of clarity; reworking this has made our paper substantially better.

---

> ### Author Response · Authors · 2025-11-22
> **Continued**
>
> __Presentation Issues (Weaknesses)__
>
> Thank you for pointing out places where our notation and wording were unclear. Sections 3.1 and 3.2 are being thoroughly revised based on this feedback and a new draft will be available shortly.  To specifically answer your questions:
>
> Section 3.1:
>
> * While you are correct that in an Ecological Game (EG), an agent would be able to observe the full state s_t and thus the identities of other agents, our environment is a Partially Observable Ecological Game (POEG), briefly mentioned at the end of 3.1.  Here, agents do not have direct access to s_t and are presented with a more restricted observation of the environment sampled from an observation function $O(s_t​, i) → p(o_{t,i} ∣ s_t​)$.  You are correct that a fully observable EG would be very unusual and different from standard evolutionary or developmental systems; we meant to describe it this way in order to draw parallels to MDPs and and POMDPs, but we regret the confusion.  As stated earlier, this subsection is being rewritten for clarity, and an updated draft will be available soon.
>
> * We appreciate the suggestion, and indeed, the sentence on line 164 is ambiguous. We will revise to: “The transition function $T$ maps the current state and the joint action vector, i.e. $(s_t, a_t)$, to a distribution over next states.”
> You are correct that $\Gamma(s_t)$ reports the list of agents alive at time $t$ and their parents.  While agents in our simulator have only a single parent, an ecological game could support multi-parent inheritance, so for full generality, parents are expressed as a list of parents for each agent.
>
> * You are correct that $\mathcal{G}_t$ should not exist.  This was an error when working from an earlier draft of this section.  We apologize and appreciate the attention to detail.
>
> Section 3.2:
>
> * Again, thank you for pointing out the notation issues. We agree that the current phrasing mixes a description of the state with a description of the state space. Your interpretation is correct, our intended meaning is:
> $s_t​=(s_t^m,s_t^a​​) \in S=S^m × S^a$
>
> * The only static component of s_t is a global terrain height.  This determines which portions of the map will be under water.  The water and distribution of resources change with time as they are consumed by the agents and move according to the environment dynamics.  Specifically, water flows downhill, so that if an agent dies on land, its water will flow back down into the sea.  All three resources can be consumed by the agents, which changes the spatial distribution of these resources over time.  The total water and biomass in the system is constant, but the total energy changes.  Spent energy is not returned to the map, but energy will grow on freestanding biomass in the map.
>
> * Presently, the agents do not have any explicit memory component and make decisions based only on immediate observations.  While this is somewhat limited, it was an intentional decision in order to limit complexity and reduce the parameter count.  Details of the agent policies can be found in Section 3.3.  Memoryless reactive policies have been shown to be effective in several environments such as Atari games.  With that said, including memory is on our road map for future work.
> Based on this feedback and that of the other reviewers, we are conducting several additional ablation experiments to quantify the sensitivity of emergent behavior to specific parameter values, including:
>     * Varying the initial population along with the map size
>     * Varying the network mutation rate
>     * Varying the network size
>     * Varying the agents’ sensory range
>     * Varying the amount of resources in the environment
>     * Varying the damage done by attacks
>     * Decreasing the agents’ initial HP so that they must grow to full health over time
>
> Again we appreciate the detailed constructive comments and apologize for the missing information.  These notes will substantially improve the paper, and we look forward to feedback on our updated draft, which will be available soon.
>
> [Baker et al. ‘19] “Emergent Tool Use From Multi-Agent Autocurricula”
>
> [Open-Ended Learning Team ‘21] “Open-Ended Learning Leads to Generally Capable Agents”
>
> [Suarez et al. ‘19] “Neural MMO: A massively multiagent game environment for training and evaluating intelligent agents”
>
> [Suarez et al. ‘20] “Neural MMO v1. 3: A massively multiagent game environment for training and evaluating neural networks”
>
> [Suarez et al. ‘23] “Neural MMO 2.0: a massively multi-task addition to massively multi-agent learning”
>
> [Lu et al. 24] “JaxLife: An Open-Ended Agentic Simulator”
>
> [Hamon et al. 23] “Eco-evolutionary dynamics of non-episodic neuroevolution in large multi-agent environments”

---

### Author Response · Authors · 2025-12-03
**Rebuttal Period Summary to New AC**

We thank the Area Chair for the opportunity to clarify our contributions and address the reviewers’ concerns. We sincerely appreciate the reviewers’ detailed and constructive feedback, which substantially improved the paper. Below, we summarize how we have addressed the issues raised across all five reviews in our updated manuscript.

**Reviewer 1**

**Main concerns**

1. Insufficient contextualization vs. prior work and discussion of contributions to state-of-the-art
2. Lack of rigor and clarity in the formalism and problem setup
3. Lack of description of simulator design and its contributions

**How we address**

1. We updated our abstract, introduction, related work, and conclusion to better contextualize our paper and make our primary contributions clearer. To clarify, our main contributions are:

   a. To our knowledge, our experiments contain some of the largest simultaneous populations of agents controlled by deep neural networks to date.

   b. This scale gives us the unprecedented ability to examine the effects of physical size and population on the emergence of complex behavior due to evolutionary response to ecological pressure.

   c. We show that increasing scale produces new emergent behavior more reliably, and demonstrate the effects of different sensor modalities on the emergence of these behaviors. We conduct extensive ablations to quantify the sensitivity of emergent behavior to specific environment and agent hyperparameter configurations in Appendix F of the updated manuscript, as well as statistical analyses of the effect of world size and compass use on emergence and extinction timing in Appendix B.

2. We substantially reworked section 3.1 to address the issues with the clarity and formalism in these sections.
3. We improved discussion of the simulator in section 3.4 and added Appendix E to provide more details.

---

**Reviewer 2**

**Main concerns**

1. Lack of clear novelty relative to previous Neural MMO and Hide-and-Seek papers
2. Unclear contribution on behavioral emergence as a result of scale
3. Uses only memoryless agents, limiting comparisons with more capable systems

**How we address**

1. We have substantially modified the abstract, introduction, related work and conclusion to make our contributions and relationship to related work more clear.  With respect to Neural MMO and Hide-and-Seek, our environment has no explicit objective (both of these prior works use reinforcement learning to explicitly train agents) and contains life, death, and reproduction rules in order to simulate population dynamics and ecological pressures.  We also emphasize the properties of emergent behaviors as a function of environmental scale, which is also studied to some extent in Neural MMO, but not Hide-and-Seek.  While these environments include more complex dynamics than ours, they are substantially smaller in terms of the number of supported agents.
2. We add a number of ablations in Appendix F to further demonstrate how scale drives emergence of complex behaviors under different environment and agent configurations. These ablations also disentangle the effects of several design decisions in our environment and policy networks. Furthermore, in Appendix B, we perform runs across many more seeds and fit a Kaplan-Meier estimator to the time-to-extinction distribution and Cox proportional hazards model comparing emergence and extinction timing, showing that worlds tend to survive longer and facilitate quicker behavioral emergence if they are large or if agents have compass information.
3. We added text to Section 3.3 justifying our choice of memoryless agents for scalability and as a way to investigate the extent to which complex behaviors can still develop due to evolutionary pressures in spite of minimal agent capabilities and network size. In future work, we intend to extend our work to memory-based agents.

---

> ### Author Response · Authors · 2025-12-03
> **Continued**
>
> ---
>
> **Reviewer 3**
>
> **Main concerns**
>
> 1. Lack of quantitative behavioral metrics
> 2. Minimal evolutionary mechanism
> 3. Missing ablations and baselines
> 4. Unclear compute requirements and reproducibility
>
> **How we address**
>
> 1. Although directly measuring behavior in such a large population is challenging, our original draft included analysis of several quantitative metrics that correspond to mining, foraging, and predation behaviors, including % drop in free biomass on land, rates of move, eat, and attack actions, as well as effectiveness of attacks.  Additionally, our new Appendix B includes statistical analyses of the rates at which the mining behavior emerges at different scales and sensor configurations.  Originally, we intended to include some cluster-based behavioral metrics, but this was not possible in time.
> 2. We added a small edit to Section 3.3 to clarify and justify our use of mutation alone as an update mechanism.  As we clarified with the reviewer, our goal was not to maximize complexity with biological realism, but to show that meaningful behavioral strategies can arise even under simple evolutionary mechanisms (mutation only), and to analyze how environmental structure shapes these outcomes.
> 3. This concern was shared across reviews. We have added extensive new ablations to evaluate sensitivity to environmental, evolutionary, and agent-level parameters in Appendix F of the updated manuscript.
> 4. We have updated section 3.4 and added Appendix E detailing hardware, runtime, non-determinism, and scaling behavior, as well as further details of our simulator.
>
> ---
>
> **Reviewer 4**
>
> **Main concerns**
>
> 1. Mutation-only neuroevolution with memory-less MLP policies
> 2. Unclear relation to prior open-ended learning frameworks Novelty Search, MAP-Elites and POET
> 3. Unclear contributions and new behavioral insights
> 4. Confounding between environmental scale and population size
>
> **How we address**
>
> 1. We added clarifying text to Section 3.3 justifying our choice of memoryless agents for scalability and the use of mutation alone for policy modification as a way to investigate the extent to which complex behaviors can still develop due to evolutionary pressures in spite of minimal agent capabilities and network size. In future work, we intend to extend our work to memory-based agents.
> 2. We have substantially modified the abstract, introduction, related work and conclusion to make our contributions and relationship to related work more clear.  As we pointed out to the reviewer,  MAP-Elites and POET are all algorithms for finding solutions to problems, where we use an open-ended framework without an explicit objective.  Like our work, Novelty Search is also not trying to find a solution to an objective.  Unlike our work, it is still trying to identify interesting candidates, whereas we are attempting to understand the relationship between the environment and emergent behavior.
> 3. As noted above, we have substantially revised the abstract, introduction, related work, and conclusion to make our contributions more clear in regards to behavioral insights. While it is intuitive that compass sensing would aid navigation and vision would enhance foraging and predation as noted by the reviewer, we show that mutation and competition-based population dynamics alone are enough to produce these behaviors.  Furthermore, we hope that demonstrating the effects of scale on the emergence of these behaviors provides useful guidance for others working in similar settings.
> 4. We have added ablations disentangling environmental scale and initial population size in Appendix F.1, showing that initial population size has little effect on consistency of emergent mining behavior or the rate of extinction, while environment scale has a significant impact. We also perform detailed statistical analyses (fitting a Cox proportional hazards model and Kaplan-Meier estimator) in Appendix B, demonstrating the effects of world size and compass use on emergence and extinction rates.

---

> > ### Author Response · Authors · 2025-12-03
> > **Continued**
> >
> > ---
> >
> > **Reviewer 5**
> >
> > **Main concerns**
> >
> > 1. Limited experiments and simulator details
> > 2. Weak statistical treatment with regard to emergent behavior, and possible self-averaging
> > 3. Unclear explanation for GPU non-determinism
> > 4. Question on agent color
> >
> > **How we address**
> >
> > 1. In the revised manuscript, we have added substantially more ablations and analyses in Appendix F, including studies on the effect of the initial population size, resource rules, mutation rates, network size, vision range, attack strength, and HP hyperparameters.  Additionally we have revised section 3.4 discussing the simulator and added Appendix E with more simulator and computational details.
> > 2. We run additional experiments to determine whether behaviors in larger environments are simply self-averaging or qualitatively different. These results are included in Appendix F.3 and indicate that larger environments enable and stabilize mining behavior in ways that cannot be explained by averaging over many small simulations. In Appendix B, we also perform runs across many more seeds and a fit Kaplan-Meier estimator to the time-to-extinction distribution, observing that worlds tend to survive longer and produce mining more quickly if they are large or if agents have compass information. In addition, we fit a Cox proportional hazards model comparing emergence and extinction timing, showing that both larger world size and compass use substantially reduce the hazard (extinction).
> > 3. We explain the source of GPU non-determinism in Appendix E.  The reviewer asked about the effects of 16-bit precision for the neural network weights, but it appears that the issue has more to do with nondeterministic ordering of operations such as sums.
> > 4. We have updated the description of the agents’ mutable color trait in Section 3.3.  The color controls what the agent looks like when viewed by another agent with a vision sensor.

---

### Meta-Review · Area_Chair_BW9a · 2026-01-08

**Summary:**

The paper introduces a large-scale JAX-based ecological simulation to investigate emergent behaviors in evolving populations without explicit rewards. The decision to reject is primarily informed by the reviewers' concerns regarding the methodological rigor and novelty of the study. Specifically, multiple reviewers noted the absence of quantitative metrics to substantiate claims of emergent complexity, relying instead on qualitative observations. There were significant concerns regarding the lack of clear differentiation from prior work like Neural MMO and POET. Furthermore, the use of memoryless agents and mutation-only evolution was criticized for limiting the potential for complex behavior, and the experimental design was found to confound environmental scale with population size, obscuring the true drivers of the observed stability.

**Reviewer Concerns:**

The reviewers raised several critical concerns that justify a rejection:
1. hMQJ and ssLs argued that the work lacks sufficient contextualization and novelty compared to prior work such as Neural MMO and Hide-and-Seek.
2. NFvD and hMQJ expressed strong concern over the lack of quantitative behavioral metrics, noting that claims of emergence rely mainly on qualitative observation.
3. ssLs and P43N highlighted that the use of memoryless MLP agents and mutation-only neuroevolution limits the complexity of learnable behaviors and interpretability.
4. P43N pointed out a confounding variable where map area and population size are scaled together, making it unclear which factor drives stability.
5. J3YU noted that stability in larger simulations might simply be the result of statistical self-averaging rather than qualitatively new dynamics.

**Reviewer Scores:**

Reviewer hMQJ: 2
Reviewer ssLs: 2
Reviewer NFvD: 6
Reviewer P43N: 4
Reviewer J3YU: 8

---

### Decision · Program_Chairs · 2026-01-26

Reject